# Targeted delivery of the probiotic *Saccharomyces boulardii* to the extracellular matrix enhances gut residence time and recovery in murine colitis

Mairead K. Heavey[1], Anthony Hazelton[1], Yuyan Wang [1], Mitzy Garner [1], Aaron C. Anselmo[1,5], Janelle C. Arthur [2,3,4] ✉ & Juliane Nguyen [1,4] ✉

Probiotic and engineered microbe-based therapeutics are an emerging class of pharmaceutical agents. They represent a promising strategy for treating various chronic and inflammatory conditions by interacting with the host immune system and/or delivering therapeutic molecules. Here, we engineered a targeted probiotic yeast platform wherein *Saccharomyces boulardii* is designed to bind to abundant extracellular matrix proteins found within inflammatory lesions of the gastrointestinal tract through tunable antibody surface display. This approach enabled an additional 24–48 h of probiotic gut residence time compared to controls and 100-fold increased probiotic concentrations within the colon in preclinical models of ulcerative colitis in female mice. As a result, pharmacodynamic parameters including colon length, colonic cytokine expression profiles, and histological inflammation scores were robustly improved and restored back to healthy levels. Overall, these studies highlight the potential for targeted microbial therapeutics as a potential oral dosage form for the treatment of inflammatory bowel diseases.

Engineered probiotic microbes represent an emerging pharmaceutical strategy with the ability to modulate the gut microbiome[1-3], deliver therapeutic payloads[4,5], support host metabolism[6-9], and interact with the host immune system to regulate the inflammatory process[10,11]. The recent FDA approval of the first microbiome-based therapy highlights both the clinical relevance of this strategy as well as the infancy of this field with regard to the unique formulation and delivery challenges of therapeutic microbes[12].

The gut microbiome has become increasingly important in understanding the pathology and progression of chronic diseases such as inflammatory bowel diseases (IBD)[13,14]. Unfortunately, IBD remains

difficult to treat, with many patients losing response to current therapeutic options and eventually requiring invasive surgery[15]. IBDs, which include Crohn's disease and ulcerative colitis, are chronic, relapsing-remitting conditions that affect more than 58,000 children and 1.2 million adults in the United States alone[16]. Many currently available drugs have severe side-effects, are not targeted to inflammatory sites within the gastrointestinal (GI) tract, and require administration via subcutaneous injection or, in the case of Remicade®, hospital visits for infusions every 8 weeks[17]. Engineered therapeutic microbes have the potential to serve as a locally acting treatment strategy, reducing the potential for systemic side effects commonly

[1]Division of Pharmacoengineering and Molecular Pharmaceutics, Eshelman School of Pharmacy, University of North Carolina at Chapel Hill, Chapel Hill, NC 27599, USA. [2]Department of Microbiology and Immunology, The University of North Carolina at Chapel Hill, Chapel Hill, NC, USA. [3]Center for Gastrointestinal Biology and Disease, The University of North Carolina at Chapel Hill, Chapel Hill, NC, USA. [4]Lineberger Comprehensive Cancer Center, The University of North Carolina at Chapel Hill, Chapel Hill, NC, USA. [5]Present address: VitaKey Incorporation, Durham, NC 27701, USA. ✉e-mail: janelle_arthur@med.unc.edu; julianen@email.unc.edu

associated with systemically administered therapeutics. Engineered microbes and probiotics are also amenable to formulation into oral dosage forms, which is a preferred administration route for patients in comparison to injection or infusion routes. Furthermore, they can be engineered to exert therapeutic effects through complementary mechanisms of action[18].

A primary and significant limitation in this field is enabling a controlled and prolonged retention time of engineered microbes within the GI tract and further targeting them to specific disease sites[19–22]. Following oral administration, microbes rapidly transit through the GI tract while displaying minimal interaction with the host tissue[23–25]. Many probiotic microbes experience a phenomenon called "colonization resistance" where the probiotic microbes are outcompeted for space and nutrients by the resident commensal microbiota, resulting in rapid clearance from the gut[26]. In our studies, and in previous reports, *Saccharomyces boulardii (S.b.)* is rapidly cleared from the gut of conventionally raised mice within 24 h post-administration[27]. Previous reports include the use of either germ-free mice or antibiotic co-administration to increase residence times for *S.b*[27,28]. Germ-free conditions or continuous broad-spectrum antibiotic use is not clinically translatable for human application, substantiating the need to explore alternative strategies for improving probiotic gut residence time. Rapid transit time limits the therapeutic capabilities of these microbes and requires frequent and continued dosing, which is non-optimal when scaling up to treatment in humans[29–31]. To address these challenges, we took inspiration from several gut-resident microbes, probiotics, as well as pathogens that express cell surface proteins called adhesins, known to facilitate specific binding between the microbe and its target cell or protein[32–34]. Adhesins are typically studied in the context of pathogen invasion into host tissue or probiotic inhibition of pathogen binding[35,36]. Adhesins also play a critical role in establishing spatial niches within the GI tract to support microbial survival and proliferation[37]. We hypothesize that "synthetic adhesins" could be attached to or expressed on the surface of probiotic microbes to facilitate binding of the microbe to diseased or inflamed areas for a more optimal delivery strategy in IBD[33].

The pathophysiological characteristics of inflammatory lesions in Crohn's disease and ulcerative colitis are unique and multifaceted. These characteristics include the infiltration of inflammatory immune cells into host tissue, an increase in the expression of inflammatory cytokines, mucosal and epithelial sloughing, and overexpression and deposition of extracellular matrix proteins[38,39]. Through the fibrotic remodeling process, as inflammatory lesions become more severe, the mucus and epithelial layer is sloughed away, exposing deeper layers such as the submucosa and lamina propria where both full-length and proteolytically degraded extracellular matrix (ECM) proteins are overexpressed and deposited, generating a dense matrix that is resistant to healing due to severe and dysregulated inflammation[40–43]. Among the ECM proteins overexpressed in ulcerated tissue and inflammatory lesions are multiple collagen types, laminin, fibronectin, vitronectin, and fibrinogen[39,41,42]. The increased deposition of ECM proteins within the inflamed gut represents a unique structural landmark to which targeted therapeutics could potentially bind[44,45].

To create yeast probiotics with prolonged gut residence time, we have developed a platform approach capable of binding to a variety of ECM protein targets. Here, we genetically engineered the probiotic yeast, *S.b.* to express monomeric streptavidin on its cell surface to function as a tunable handle for the attachment of biotinylated binding moieties specific to the ECM. We chose *S.b.* as the chassis organism in this application for several reasons. First, *S.b.* has clinical relevance in IBD as it is used as a supplement for human patients due to its intrinsic anti-inflammatory properties and can significantly improve inflammation when continuously dosed in preclinical models of colitis[23,46–48]. Further, *S.b.* is: 1) a genetically tractable organism with a defined

genetic engineering toolkit amenable to yeast surface display approaches[27], 2) a eukaryotic organism capable of recombinant protein production with proper folding and post-translational modifications[49], 3) amenable to auxotrophic selection, ensuring microbial biocontainment and removing the necessity for antibiotic resistance genes[50], and 4) amenable to removal through anti-fungal administration with reduced impact on the gut microbiome as compared to broad-spectrum antibiotics[51,52].

In this work we characterize the probiotic phenotype of native and engineered *S.b.*, as well as the temporal dynamics of targeting ligand retention and function. We demonstrate that *S.b.* targeted to fibronectin or collagen IV results in prolonged gut residence time, leading to substantial improvements in therapeutic efficacy in acute and chronic models of murine colitis. Overall, this work integrates principles of pharmacoengineering, microbiology, immunology, and genetic engineering to create a probiotic drug delivery approach capable of improving the delivery of engineered microbes to the gut for enhanced efficacy and improved patient-friendliness of live biotherapeutic products.

## Results

### Engineering the *S. boulardii* cell surface to enable targeting to the extracellular matrix

Rapid and untargeted transit through the GI tract results in minimal host-microbe interactions, thus decreasing the opportunity for orally administered microbes to exert anti-inflammatory, probiotic effects on the host[34]. To enable a longer probiotic gut residence time within the colon, we engineered a platform surface display system (Fig. 1A) to confer targeting of *S.b.* to extracellular matrix proteins upregulated in ulcerations and inflammatory lesions of mouse models and human ulcerative colitis[53–59]. First, we genetically engineered *S.b.* to stably display monomeric streptavidin (mSA) on its cell surface. To achieve this, we created an integrative plasmid to confer a constitutive expression of the yeast cell surface a-agglutinin anchorage subunit AGA1, as well as constitutive expression of its binding partner subunit, AGA2 fused to mSA (Fig. 1B and Supplementary Fig. 1)[60,61]. The mSA protein sequence has been previously validated to bind to biotin with high specificity and reproducibility[62,63]. Auxotrophic markers were used to enable the selection of positively transformed colonies without the use of antibiotic-resistance genes. Genomic integration of the surface display plasmid allows for continuous expression of the engineered construct without the need for selective media exposure, an important feature for use as therapeutic in the GI tract. The *S.b.* strain engineered to express mSA on the cell surface has been termed, *S.b. mSA*.

We performed binding assays and flow cytometric analysis with several biotinylated antibodies. Titrating concentrations of fluorescently labeled *S.b.* or *S.b. mSA* were added to biotin-coated well plates. We then used ImageJ to quantify the number of bound cells per image. Our results showed that *S.b. mSA* bound to the biotin-coated well plate in a concentration-dependent manner, as evidenced by qualitative fluorescent images (Fig. 1C) and quantitative image analysis (Fig. 1D). In contrast, *S.b.* was unable to effectively bind to the biotin-coated well plate. Next, we investigated whether biotinylated antibodies could attach to the cell surface via streptavidin-biotin interactions. *S.b. mSA* was incubated with titrating concentrations of either biotinylated or non-biotinylated antibodies (e.g., anti-fibronectin, anti-fibrinogen, or anti-collagen IV). Bound antibodies were then fluorescently labeled, and the binding affinity was determined with flow cytometry. We found that *S.b. mSA* binds to various biotinylated antibodies with a binding affinity ($K_D$) between 1.11–1.47 nM (Fig. 1E). Conversely, no fluorescent signal was detected when *S.b. mSA* was incubated with non-biotinylated antibodies. These findings indicate that *S.b. mSA* can be labeled with a variety of biotinylated antibodies through specific and high affinity streptavidin-biotin interactions.

 

## Enabling *S. boulardii* to selectively bind to the extracellular matrix

A hallmark of chronic inflammatory lesions in ulcerative colitis and Crohn's disease, and a proposed pre-symptomatic biomarker, is increased extracellular matrix protein deposition in colonic tissue[39–42,44,45]. These proteins can be found either proteolytically degraded or in their functional form in high abundances within the mucosal and submucosal layers. We propose that the overabundant ECM proteins can serve as a viable target to increase probiotic retention within the inflamed GI tract. We characterized binding capabilities of the engineered yeast to three different ECM proteins:

1) fibronectin, 2) fibrinogen, and 3) collagen IV. A comprehensive analysis of 7 whole-genome transcriptional analysis datasets from colonic biopsies of healthy ($n = 82$) and active ulcerative colitis human subjects ($n = 181$) reveals a significant increase in expression of these proteins in the colonic mucosa of human subjects with ulcerative colitis (Fig. 1F)[53–59], further validating the clinical relevance and the selection of these target proteins. To evaluate the binding capabilities, *S.b. mSA* was fluorescently labeled and incubated with biotinylated antibodies against fibronectin (FN), fibrinogen (FB), or collagen type IV (CIV) (Fig. 2A). The antibody labeled *S.b.* strains were termed, *S.b. FN, S.b. FB, or S.b. CIV*, respectively, and

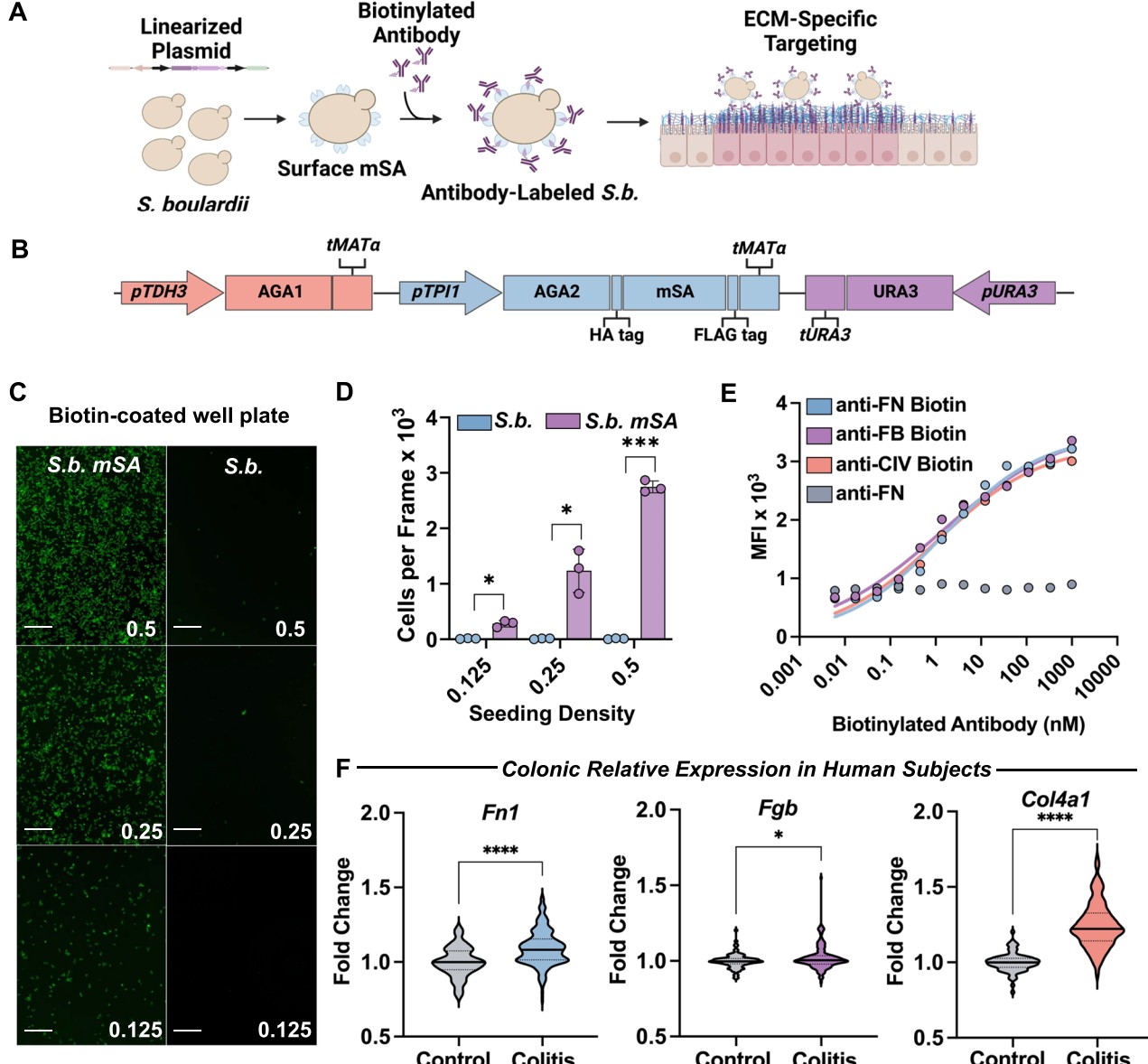

**Fig. 1 | Genetic engineering of *S. boulardii* to express *mSA* handle on the cell surface for attachment of ECM-specific targeting ligands. A** Schematic of engineering steps to enable the binding of *S.b.* to extracellular matrix proteins. **B** Schematic of key components of plasmid used to engineer stable expression of monomeric streptavidin (mSA) on the yeast surface. **C** Representative fluorescent images of *S.b. mSA* (left) and *S.b.* (right) attached to a biotin-coated well plate incubated at varying initial cell concentrations (starting $OD_{600} = 0.5$, 0.25, or 0.125). Scale bars are 100 μm. **D** Quantification of attached *S.b.* on the biotin-coated well plate. Images were quantified using ImageJ, $n = 3$ wells per condition. Data are shown as mean ± SD. Significance was determined using unpaired, two-tailed Student's *t*-tests, $n = 3$ per condition. **E** Mean fluorescence intensity (MFI) values

resulting from labeling *S.b. mSA* with titrating concentrations of biotinylated antibodies against fibronectin (FN, blue), fibrinogen (FB, purple), or collagen IV (CIV, red) or non-biotinylated antibodies (grey) as measured by flow cytometry ($n = 3$, points represent mean). **F** Collection of fold change values in colonic expression of fibronectin (*Fn1*), fibrinogen (*Fgb*), or collagen IV (*Col4a1*) in human subjects with active ulcerative colitis ($n = 181$) as compared to healthy subjects ($n = 82$) from 7 whole-genome transcriptional analysis datasets. For violin plots, the mean is represented by the solid line and interquartile ranges represented by dotted lines. Significance assessed by unpaired, two-tailed Student's *t*-tests. $\alpha = 0.05$, *$p < 0.05$, **$p < 0.01$, ****$p < 0.0001$.

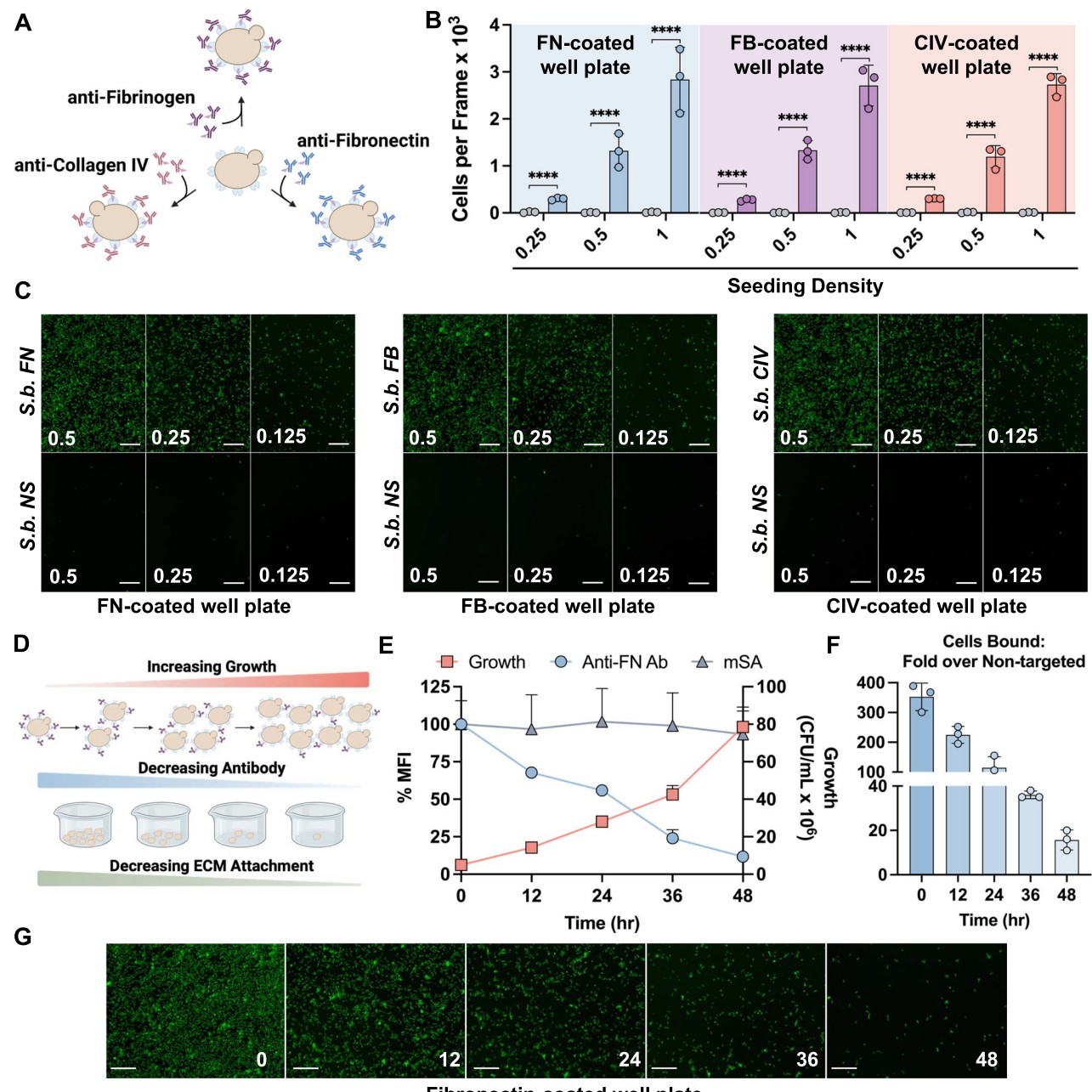

**Fig. 2 | Antibody-labeling enables *S. boulardii* to bind to corresponding extracellular matrix proteins. A** Schematic of biotinylated antibodies specific to extracellular matrix proteins attached to the *S. boulardii* cell surface via biotin-streptavidin interactions. **B** Quantification of attached targeted *S.b.* (colored bars) or non-targeted *S.b.* (grey bars) on fibronectin-coated (FN blue), fibrinogen-coated (FB purple), or collagen IV-coated (CIV red) well plates at varying seeding densities. Images were quantified using ImageJ, *n* = 3 wells per condition. Data are shown as mean ± SD. Significance was determined using ordinary one-way ANOVA with Šídák's multiple comparisons test, *n* = 3 per condition. **C** Representative fluorescent images of Left: *S.b. FN* or non-specific antibody-labeled *S.b.* (*S.b. NS*) attached to a fibronectin-coated well plate; Middle: *S.b. FB* or *S.b. NS* attached to a fibrinogen-coated well plate; Right: *S.b. CIV* or *S.b. NS* attached to a collagen IV-coated well plate. Scale bars are 100 µm. **D** Schematic depicting the growth of antibody-labeled

*S. boulardii*, resulting in the dilution of the number of antibodies remaining on each cell surface, and subsequent reduction in the capacity to bind to the corresponding ECM protein. **E** Anti-fibronectin antibody-labeled *S. boulardii* growth (red squares, right y-axis), percent antibody remaining on the cell surface as determined by flow cytometry (blue circles, left y-axis), and percent mSA remaining on the cell surface as determined by flow cytometry (grey triangles, left y-axis) over time incubated in supplemented simulated intestinal fluid (*n* = 3 technical replicates, points represent mean). Data are shown as mean ± SD, *n* = 3. **F** Quantification of plate-bound *S.b. FN* as compared to non-targeted *S.b.* on a fibronectin-coated well plate following incubation in supplemented simulated intestinal fluid (Images were quantified using ImageJ, *n* = 3 wells per condition) Data are shown as mean ± SD. **G** Representative fluorescent images of *S.b. FN* bound to a fibronectin-coated well plate over time. Scale bars are 100 µm. α = 0.05, ****$p < 0.0001$.

were added to corresponding protein-coated well plates. For each condition we included controls composed of *S.b. mSA* labeled with biotinylated antibodies not specific (NS) to the coated ECM protein (*S.b. NS*). Bound cells were imaged via fluorescent microscopy and the number of cells bound quantified by ImageJ. We found

that *S.b. mSA* labeled with biotinylated antibodies displayed concentration-dependent binding to the corresponding ECM proteins (Figs. 2B, C). However, when *S.b. mSA* was labeled with antibodies not specific to the ECM protein, minimal to no probiotic binding was observed.

To further explore the temporal dynamics of cell surface ligand retention and ECM binding capacity as it relates to growth and dilution in the in vivo environment, we labeled *S.b. mSA* with biotinylated anti-fibronectin (FN) antibodies then subjected them to culture in modified simulated intestinal fluid (Fig. 2D). Over the course of 48 h, we measured 1) *S.b.* growth, 2) amount of biotinylated antibody remaining on the cell surface, 3) expression of mSA on the cell surface, and 4) capability of *S.b.* to bind to fibronectin (Fig. 2E). Over 48 h, we observed a slow but exponential increase of *S.b.* growth, as well as an exponential decrease in the mean fluorescence intensity corresponding to the presence of biotinylated anti-FN on the cell surface. The fluorescent signal corresponding to mSA expression on the cell surface remained constant throughout the entirety of the study, suggesting that the decrease in antibody signal over time is not due to a decrease in mSA expression. The decrease in anti-FN fluorescent signal over time suggests that as *S.b.* grow and divide, the amount of antibody remaining per cell decreases (Figs. 2F, G). However, even after 48 h of culture, incubation of *S.b. FN* on a fibronectin-coated well plate still resulted in a ~20-fold increase in the number of plate-bound cells as compared to incubation with non-targeted *S.b.*. The dynamics explored here can provide insights into the in vivo pharmacokinetics and clearance mechanisms of engineered *S.b.*

## Genetically engineered *S. boulardii* retains probiotic phenotype

*S.b.* is capable of surviving the harsh environment of the early GI tract, modulating host immune response, and stimulating resident microbes to produce short-chain fatty acids (SCFA)[46–48]. These characteristics make *S.b.* an ideal candidate for further engineering and drug delivery applications for inflammatory diseases. However, genetic engineering can occasionally alter phenotypic characteristics. Thus, we evaluated these phenotypic characteristics of the non-probiotic yeast strain *Saccharomyces cerevisiae* (*S.c.*), wild type *S.b.*, and engineered *S.b.* (Fig. 3A).

The yeast strains were cultured under conditions representative of the stomach and early GI tract, and viability after culture was determined[64]. Culture conditions included media at pH 2.5, pH 4.0, 0.3% OxGall bile acids, or 0.6% OxGall bile acids. Percent viability was determined as compared to the resultant viability when each yeast strain was cultured without any physiological challenges. Under all conditions, the non-probiotic yeast strain *S.c.* exhibited the lowest end-point viability, as expected. Conversely, under all challenge conditions *S.b.*, *S.b. mSA*, and *S.b. FN* exhibited no significant deficits in viability as compared to the non-probiotic strain (Fig. 3B). Although the viability of all strains was decreased when physiological challenges were present, all *S.b.* strains outperformed the non-probiotic strain under all conditions tested.

Short-chain fatty acids are molecules produced by probiotic *S. boulardii* which are known to have potent anti-inflammatory effects on the host[65]. To evaluate whether genetic modification and/or the presence of antibodies impact the secretion of commonly measured short-chain fatty acids, we measured the concentration of acetate, propionate, and butyrate in the supernatant of *S. boulardii* cultures as a function of time, at 6 h, 12 h, and 18 h. Figure 3C, D illustrates that all groups exhibit comparable secretion levels of propionate, with a minor difference (1.1-fold) between the *S.b. FN* and unmodified S.b. Regarding butyrate expression, we observed slightly enhanced secretion compared to unmodified *S.b.*, however, the enhancement in secretion rate was smaller than 1.5-fold. Similarly, the difference we observed in acetate expression between *S.b.* and *S.b. mSA* was minimal (<1.2-fold). Overall, our findings suggest that genetic modifications and the presence of antibodies did not reduce the ability of engineered *S.b.* to secrete short-chain fatty acids (Fig. 3C, D).

*S.b.* can modulate host immune cells to exhibit an anti-inflammatory phenotype, contributing to its beneficial use in controlling intestinal inflammation[66–68]. We explored the capability of

*S.b.* to stimulate IL-10 expression by co-incubating murine bone marrow-derived dendritic cells (BMDC) ($3 \times 10^5$ cells/well) with the various yeast strains ($1 \times 10^5$ yeast cells/well) and measuring secretion of IL-10 (Fig. 3E). Measurement of IL-10 production in BMDCs was chosen as an additional assay because several reports indicate that co-culture of BMDCs with *S. boulardii* stimulates the production of IL-10[69,70]. IL-10 plays a crucial role as an anti-inflammatory cytokine in the progression of IBD. This significance has been observed not only in pre-clinical models but also in human disease progression. IL-10 acts as a natural regulator, dampening the acute inflammatory processes associated with IBD[71–73]. Co-incubation with the non-probiotic *S. cerevisiae* did not induce IL-10 production, similar to the PBS-treated negative control. In contrast, co-incubation with *S.b.* and the engineered variants significantly induced IL-10 production, with no difference in IL-10 induction between wild type *S.b.* vs. engineered *S.b.* Altogether, these results indicate that our genetically engineered *S.b.* maintains its probiotic phenotype similar to that of the wild type strain. Our findings suggest that genetic engineering does not compromise the probiotic properties of *S. boulardii*.

Inhibition of the proinflammatory NFκB pathway represents another mechanism in which *S. boulardii* exerts an anti-inflammatory effect in the host[74–78]. It has been reported that the co-culture of *S. boulardii* with TNFa-stimulated HT-29 intestinal epithelial cells inhibits the production of IL-8 through the NFκB pathway[74,79]. To evaluate this mechanism, we performed a co-culture assay of HT-29 cells stimulated with TNFa (20 mg/ml) in combination with the various *S.b.* strains or with PBS control. We measured the concentrations of IL-8 in the co-culture supernatants, a cytokine regulated by the proinflammatory NFκB pathway[5,66,80]. These results show that through co-culture of the stimulated intestinal epithelial cells with any of the *S.b.* strains results in decreased concentrations of IL-8 in the supernatants as compared to monocultures of stimulated intestinal cells or co-cultures with the non-probiotic yeast strain, *S.c* (Fig. 3F). These results represent another mechanism of action of the probiotic *S.b.*, and highlight that the genetic or biochemical changes to *S.b.* in these studies do not inhibit this mechanism of action. We acknowledge that the current literature demonstrates a wide variety of mechanisms in which *S. boulardii* exerts an anti-inflammatory effect[23,48]. Here, we perform a panel of representative assays to assess key mechanisms of *S.b.* with the primary purpose of demonstrating that our genetic and biochemical modifications do not alter these mechanisms in *S.b.*

## Extracellular matrix targeting increases gut residence time of *S. boulardii* and decreases inflammation in an acute DSS-induced model of murine colitis

It has been widely reported that repeated oral administration of *S.b.* in preclinical models of IBD decreases inflammation[23,46–48,66,67,81]. However, in most reported studies, *S.b.* is administered on a daily basis in order to maintain constant exposure to the probiotic over the course of treatment. To explore the pharmacokinetics (PK) and pharmacodynamics (PD) of ECM protein-targeted *S.b.*, we established a dextran sulfate sodium (DSS)-induced model of acute colitis in mice. Briefly, mice received DSS in the drinking water for 5 days followed by a recovery phase where mice consumed normal drinking water for 3 days (Fig. 4A). A single oral gavage of $10^9$ colony forming units (CFU) of either non-targeted (*S.b. mSA*) or ECM-targeted (*S.b. FB, S.b. FN*, or *S.b. CIV*) yeast was administered on day 5. Fecal samples were collected daily following oral gavage to measure yeast concentrations in the feces over time, providing a non-invasive method to identify PK trends[22,82,83]. Both non-targeted and all ECM-targeted *S.b.* were detectable at high concentrations within the first 24 h post-gavage (Fig. 4B). Over time, *S.b. FN* remained detectable in all five mice for at least 72 h post-gavage, at significantly higher concentrations than all other groups. Viable *S.b. FN* was also detected at high amounts colonizing the colon tissue of all 5 treated mice, with a 100-fold higher *S.b.*

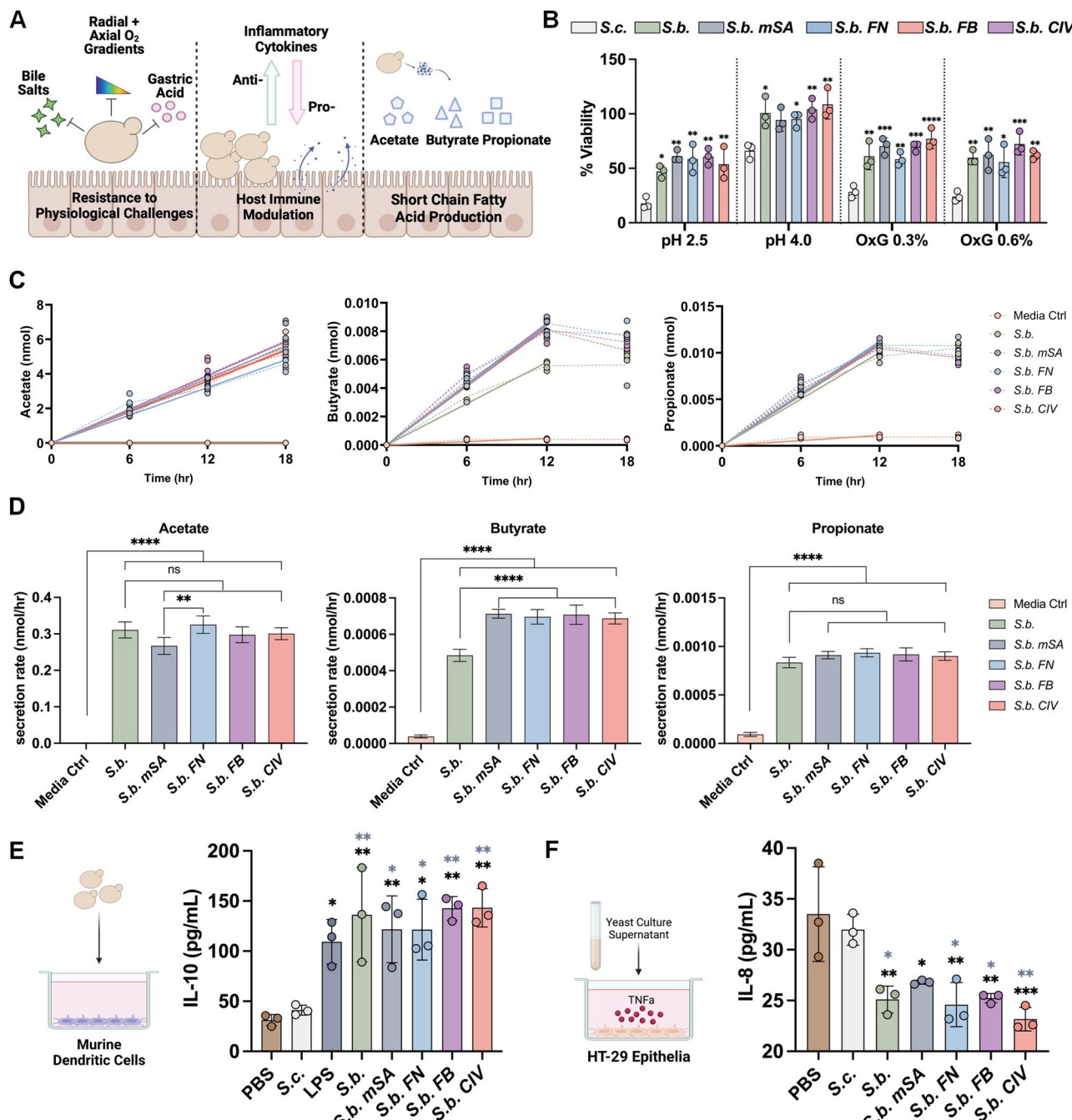

**Fig. 3 | Engineered *S. boulardii* retain probiotic mechanisms of action in vitro.**
**A** Schematic of probiotic mechanisms of action elicited by *S. boulardii*. **B** Percent
viability after 4 h incubations of control and engineered yeast in media at pH 2.5, pH
4.0, or containing 0.3% or 0.6% OxGall bile salts (OxG). Data are shown as mean ±
SD, *n* = 3 biological replicates. **C** Secretion of acetate, propionate, and butyrate
from cultures of either *S.b., S.b. mSA, S.b. FN, S.b. FB*, and *S.b. CIV*, or blank media,
*n* = 4. Solid line: simple regression of the linear portion before saturation (0 to 18 h
for acetate, 0 to 12 h for butyrate and propionate). **D** Secretion rate of acetate,
propionate, and butyrate from cultures of either *S.b., S.b. mSA, S.b. FN, S.b. FB*, and
*S.b. CIV*, or blank media calculated from linear regression in (C), *n* = 4 biological

replicates. **E** Concentration of murine interleukin-10 (IL-10) in the cell culture
supernatant following an 18 h co-incubation of murine bone marrow-derived den-
dritic cells with controls (PBS or LPS 1 μg/mL) and engineered yeast, *n* = 3 biological
replicates. **F.** IL-8 concentrations in HT-29 human epithelial cells co-cultured with
probiotic yeast strains then stimulated with TNFa (20 mg/ml). Data are shown as
mean ± SD, *n* = 3 biological replicates. Significance was determined using ordinary
one-way ANOVA with Tukey's for panels B, D, E, and F. For panel B, black asterisks
indicate significance against *S.c.*. For panels **E**, and **F**: black asterisks indicate sig-
nificance against PBS and grey asterisks indicate significance against *S.c.*, α = 0.05,
*$p < 0.05$, **$p < 0.01$, ***$p < 0.001$, ****$p < 0.0001$.

tissue concentration (Fig. 4C). These results signify that targeting *S.b.*
to fibronectin via cell surface engineering enables an extended resi-
dence time and a higher subject-to-subject pharmacokinetic con-
sistency in an acute DSS model of murine colitis. In an acute model of
DSS colitis wherein mice received DSS in the drinking water for 5 days,
dosed with *S.b. FN*, followed by an extended recovery period of

14 days; *S.b. FN* was not detectable in the feces 48 h post-dose through
end of the study at day 14 (Supplementary Fig. 3). This indicates that
after *S.b. FN* is cleared from the intestines, there is no permanent
engraftment detected (Supplementary Fig. 3).

To assess the PD effects of either non-targeted or ECM-targeted
*S.b.* in this model, we evaluated several clinical biomarkers of disease

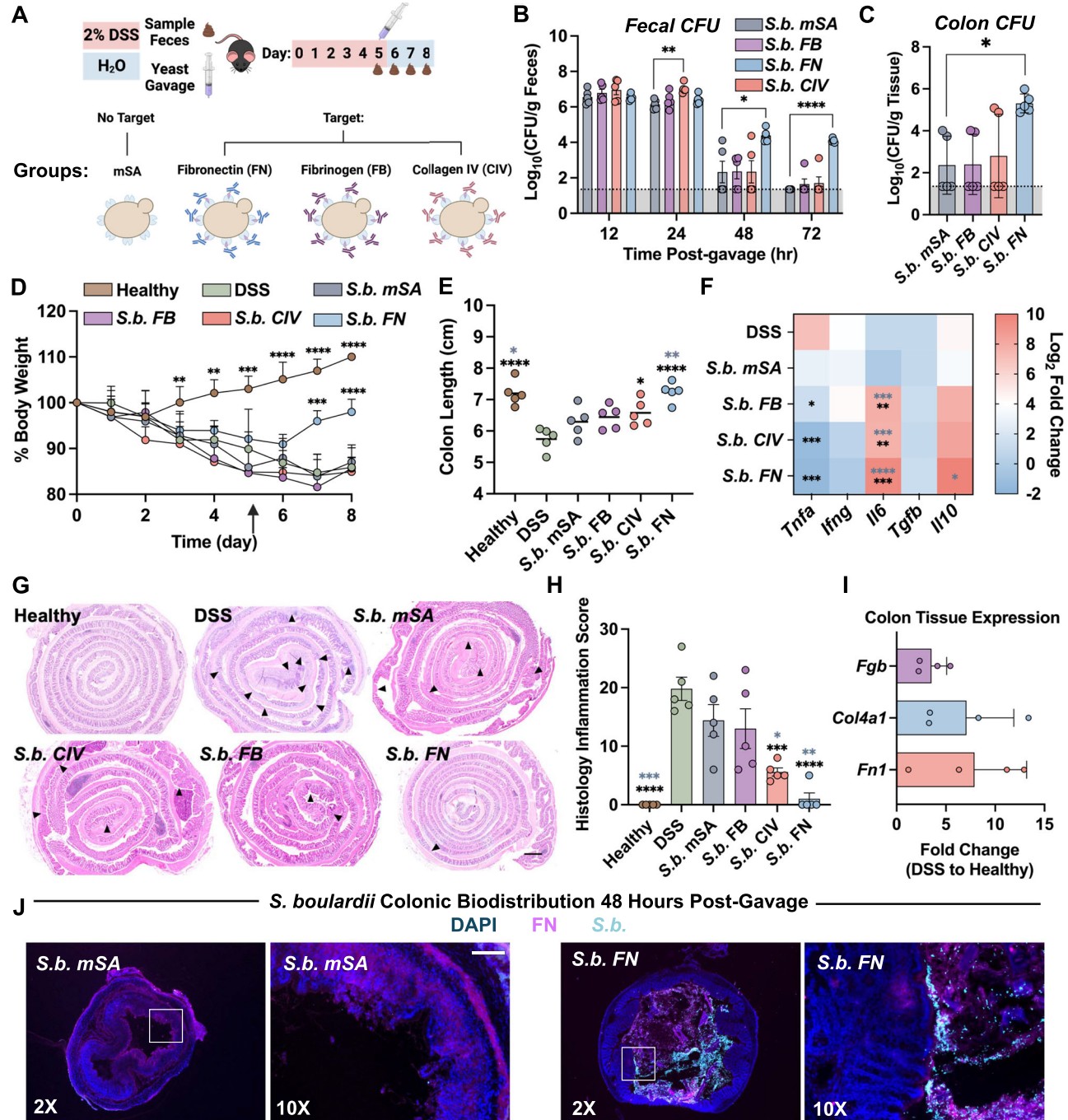

**Fig. 4 | Extracellular matrix targeting increases colon residence time of *S. boulardii* and decreases inflammation in acute DSS-induced murine colitis.**
**A** Schematic depicting time course of in vivo murine colitis study. C57BL6/J mice were administered 2% dextran sulfate sodium (DSS) in the drinking water for 5 days, then administered regular drinking water for the remainder of the study. Mice were dosed on day 5 with $10^9$ colony-forming units (CFU) of nontargeted (*S.b. mSA*), fibrinogen-targeted (*S.b. FB*), collagen IV-targeted (*S.b. CIV*), or fibronectin-targeted (*S.b. FN*) yeast via oral gavage. Stool was collected at 12, 24, 48, 72 h post-gavage to measure viable yeast by quantitative culture. **B** *S. boulardii* CFU in the stool post-gavage, n = 5 independent animals. **C** *S. boulardii* CFU in colon tissue 72 h post-gavage, n = 5 independent animals. **D** Mean percent body weight of mice over the course of the study. N = 5, significance indicates comparisons to average weights of DSS-only mice at each timepoint. Arrow indicates day of DSS removal and yeast dosing. **E** Mouse colon length at study termination, n = 5 independent animals, lines represent the mean. Black asterisks indicate significance against DSS and grey asterisks indicate significance against *S.b. mSA*. **F** Mean relative expression of pro-inflammatory (*Tnfa, Ifng, Il6*) and anti-inflammatory (*Tgfb, Il10*) cytokines in distal colon tissue compared to healthy controls, n = 5. Statistical tests between groups

for each gene were independently performed then represented in a single heatmap. Black asterisks indicate significance against DSS and grey asterisks indicate significance against *S.b. mSA*. Significance indicated as compared to mean relative expression values of DSS-only mice. **G** Representative images of hematoxylin and eosin staining of colon Swiss rolls; black arrows indicate lesions of severe inflammation/ulceration. **H** Semi-quantitative histological scores of inflammation accounting for extent of mucosal loss, hyperplasia, and erosions, n = 5 independent animals. Black asterisks indicate significance against DSS and grey asterisks indicate significance against *S.b. mSA*. **I** Mean relative expression of *Fn1*, *Fgb*, and *Col4a1* in the distal colon of DSS mice relative to healthy controls, n = 4. No statistical significance was found. **J** Representative immunofluorescence of DSS-only colons dosed with either *S.b. mSA* or *S.b. FN*. Cell nuclei (blue), fibronectin (pink), and *S.b.* (cyan) imaged with 2x and 10x objectives, n = 3. Dotted lines represent the limit of detection (Panels B and C). Data are shown as mean ± SD. Significance was determined using ordinary one-way ANOVA with Dunnett's (panels **B, C, D**) or Tukey's (panels **E, F, H**) multiple comparisons test α = 0.05, *p < 0.05, **p < 0.01, ***p < 0.001, ****p < 0.0001. Scale bars are 100 μm.

severity. Body weight decreased in all mice administered DSS, as expected, with a trend towards weight recovery in the group treated with *S.b. FN* (Fig. 4D). Compared to DSS-only mice, all groups treated with ECM-targeted *S.b.* exhibited longer colon lengths, an indicator of improved clinical disease in this model (Fig. 4E). Mice treated with *S.b. FN*, showed the most significant increase in colon length and these measurements were insignificant from those of the non-DSS healthy mice. Further analysis of colonic cytokine expression levels reveals treatment with ECM-targeted *S.b.* results in significant decreases in the expression of the inflammatory cytokine, TNFα; with *S.b. FN* treatment resulting in the most significant decrease at ~3-fold lower expression compared to DSS-only mice (Fig. 4F). Further, there was a ~1000-fold increase in the colonic expression of the anti-inflammatory cytokine, IL-10, in mice treated with *S.b. FN* as compared to DSS-only mice. Colon tissues were histologically scored for disease severity by a board-certified pathologist using a validated scoring system to evaluate mucosal loss, hyperplasia, inflammation, and extent (Figs. 4G, H)[84,85]. Mice treated with *S.b. FN* had the lowest histopathological scores of all treatment groups. Relative mRNA expression levels for *Fn1* (fibronectin), *Fgb* (fibrinogen), and *Col4a1* (collagen IV) were measured in distal colon tissues from healthy and DSS-only mice that were not administered any *S.b.* (Fig. 4I). Although not statistically significant from the other groups, fibronectin expression was the most upregulated in DSS-only colons vs. healthy colon tissue, indicating a potential trend between ECM expression and targeted probiotic retention time. We hypothesize this upregulation of *Fn1* may enable retention of the FN-targeted yeast (*S.b. FN*) in the diseased colon and promotes closer interaction between the probiotic yeast and the host tissue.

To ensure that therapeutic effects were driven by *S.b.* and not the attached antibodies, we conducted an additional study by dosing the antibodies alone. In an acute DSS model, when antibodies were orally administered alone, there were no significant differences in the colon length between control and antibody-treated mice, indicating that the antibodies only mediate binding and retention but do not exert any therapeutic effects (Supplementary Fig. 4E).

Next, we performed immunofluorescent staining to visualize the biodistribution of *S.b. mSA* and *S.b. FN* in the inflamed colon. After exposure to DSS for 5 days, as above, mice were orally gavaged with $10^9$ CFU of either *S.b. mSA* or *S.b. FN* (n = 3), and mucus-preserved colon tissues were harvested 48 h post-gavage. Tissues were stained with antibodies against fibronectin (pink) and *S.b.* (cyan), and nuclei visualized with DAPI (blue). *S.b. FN* is co-localized with fibronectin deposits within the mucosal layer lining the epithelium, whereas minimal *S.b. mSA* could be seen (Fig. 4J). These results are consistent with higher fecal loads of *S.b.* in mice treated with *S.b. FN* vs. *S.b. mSA* or *S.b. FN* (Fig. 4B). The image analysis performed here represents a qualitative snapshot showing the enhanced retention of *S.b. FN* compared to *S.b. mSA* within the colon tissue. These findings are further supported by the quantitative evaluation of colon tissue load as described in Fig. 4C.

### *S. boulardii* targeted to collagen IV increases probiotic gut residence time and reduces inflammation in a chronic DSS-induced murine model of colitis

Relapsing and remitting conditions are commonly observed in the clinical presentation of ulcerative colitis, a chronic disease where patients have cycling periods of severe inflammation followed by periods of recovery[15,17]. To better model the human disease, we established a chronic DSS-induced murine model of colitis. Utilization of repeated cycles of 3–7 days of DSS administration followed by 5–14 days of recovery without DSS is a widely reported strategy to assess chronic colitis in mice[86–90]. Here, mice received DSS in the drinking water for 5 days followed by a recovery phase where they were placed back onto normal drinking water for 3 days. This process

was repeated twice more (Fig. 5A). Mice received oral gavages of non-targeted (*S.b. mSA*) or ECM-targeted (*S.b. CIV* or *S.b. FN*) yeast every 3 days starting on day 9. Dosing with yeast commenced on day 9 of the study to allow for 1 complete cycle of inflammation and recovery. This is to resemble the human disease more closely, wherein a patient may not receive treatment until after they've gone through a period(s) of inflammation, recovery, and relapse[86,91]. Here, we combine fecal CFU concentration values from each 48- or 72-h post-dose time point across the entire study to gather a high-level snapshot of CFU concentrations in the feces (Fig. 5B). A data plot showing all the fecal CFU concentrations across the study can be found in Supplementary Fig. 5.

Fecal samples were collected 48- and 72- h post-dose to measure viable yeast by quantitative culture. Targeting *S.b.* to collagen IV (*S.b. CIV*) resulted in the highest fecal concentration, compared to non-targeted *S.b.* or FN-targeted *S.b.* (Fig. 5B and Supplementary Fig. 5). Additional analysis revealed that mice treated with *S.b. CIV* had significantly greater overall exposure throughout the study, as demonstrated by the area under the curve (AUC) of the fecal CFU versus time plot (Fig. 5C). Viable *S.b. CIV* yeast were also detected in higher abundance in the colon tissue at the study endpoint, where mice treated with *S.b. CIV* had up to 1000-fold higher concentrations of tissue-associated *S.b.* compared to other treatment groups (Fig. 5D). We observed higher subject-to-subject pharmacokinetic consistency in this chronic model, which demonstrated enhanced gut residence time by targeting *S.b.* to collagen IV.

To assess the therapeutic efficacy of the targeted *S.b.*, we measured several pharmacodynamic parameters that indicate disease severity in this model. As expected, an initial decrease in body weight was observed for all mice receiving DSS. However, mice treated with *S.b. CIV* demonstrated a significant recovery in body weight beginning after the first probiotic dose, indicating its potent therapeutic effect (Fig. 5E). Colons from mice treated with *S.b. CIV* were significantly longer than those of DSS-only mice, similar to those of healthy mice (Fig. 5F). Cytokine expression in the colon tissue further demonstrated the enhanced anti-inflammatory effects of *S.b. CIV*. Transcription of the proinflammatory cytokines *Tnfa, Ifng*, and *Il6* were significantly reduced, with increased expression of anti-inflammatory *Il10* and *Tgfb* compared to DSS-only mice (Fig. 5G). Histological scoring of colonic tissue revealed results consistent with the clinical and molecular analyses, where mice treated with *S.b. CIV* had significantly lower inflammatory scores compared to DSS-only mice (Fig. 5I). Colonic expression of *Fgb*, *Fn1*, and *Col4a1* was elevated by chronic DSS administration, with the highest relative expression of *Col4a1* (Fig. 5J). Together our data demonstrate that targeting *S.b.* to ECM that is abundant in the chronically inflamed colon enhances gut retention of this therapeutic and its anti-inflammatory effects.

## Discussion

This study presents a platform approach for targeting *S. boulardii* to extracellular matrix proteins abundant in the inflamed colon, leading to substantially improved probiotic efficacy in acute and chronic preclinical models of murine colitis. Here, stable yeast surface display of monomeric streptavidin enables a functional "handle" for the tunable attachment of a variety of targeting ligands to the cell surface. Attachment of biotinylated antibodies specific for clinically relevant ECM proteins including fibronectin, fibrinogen, and collagen IV confer *S.b.* the capability to specifically bind to the corresponding ECM protein. Binding capabilities to the ECM are increased 350-fold over non-targeted *S.b.* upon initial surface modification and are maintained at least 15-fold over non-targeted *S.b.* after 48 h in a simulated gastro-intestinal environment in vitro. Importantly, genetically, and biochemically modified *S.b.* maintained its probiotic phenotype, including resistance to gastrointestinal physiological challenges, secretion of short-chain fatty acids, and stimulation of immune cell anti-inflammatory cytokine secretion.

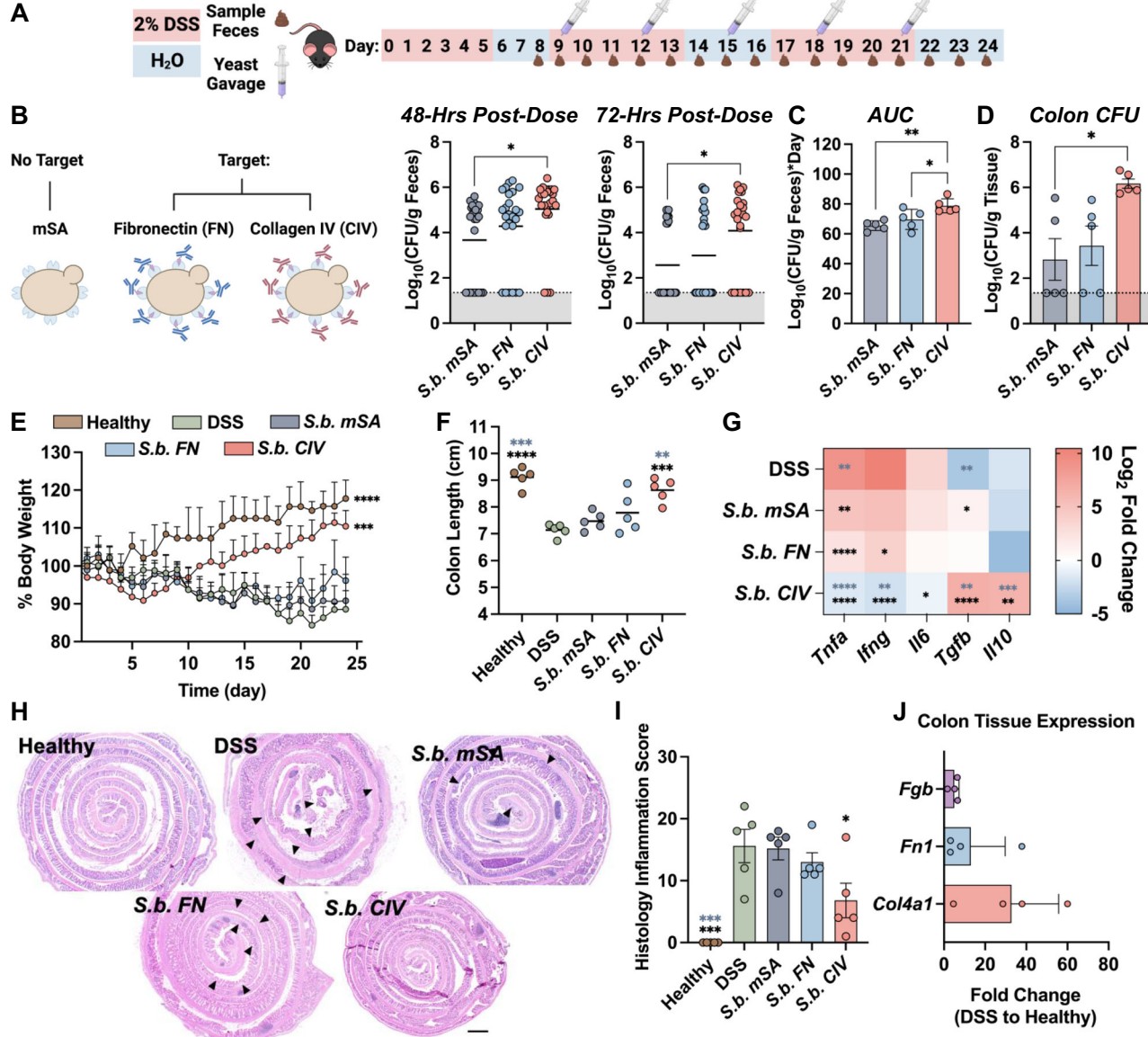

**Fig. 5 | *S. boulardii* targeted to collagen IV exhibits enhanced gut retention time and anti-inflammatory effects in a chronic DSS model of colitis. A** Schematic depicting the time course of in vivo murine colitis study. C57BL6/J mice were administered 2% dextran sulfate sodium (DSS) in the drinking water for 5 days (injury) followed by administration of regular drinking water for 3 days (recovery). This cycle was repeated twice more. Following the first cycle, mice were administered $10^9$ colony-forming units (CFU) of nontargeted (*S.b. mSA*), fibronectin-targeted (*S.b. FN*), or collagen IV-targeted (*S.b. CIV*) yeast via oral gavage every 3 days for the remainder of the study. The stool was collected every 24 h post-gavage to measure viable yeast. **B** *S. boulardii* CFU in the stool on the 48-h post-dose timepoints (Days 11, 14, 17, 20, and 23), n = 25. and on the 72-h post-dose timepoints (Days 12, 15, 18, 21, and 24), *n* = 25, lines represent mean, (5 independent animals x 5 collection timepoints). **C** Area under the curve (AUC) analysis of CFU in the stool over time plots, *n* = 5 independent animals. **D** *S. boulardii* CFU in the colon tissue at study termination, *n* = 5 independent animals. **E** Mean percent body weight of mice over the course of the study. *n* = 5, significance determined by comparing the area under each curve as compared to that of DSS mice. **F** Mouse colon length at study

termination, *n* = 5, lines represent the mean. Black asterisks indicate significance against DSS and grey asterisks indicate significance against *S.b. mSA*. **G** Mean relative expression of pro-inflammatory (*Tnfa, Ifng, Il6*) and anti-inflammatory (*Tgfb, Il10*) cytokines in distal colon tissue compared to healthy controls, *n* = 5. Black asterisks indicate significance against DSS and grey asterisks indicate significance against *S.b. mSA*. **H** Representative images of hematoxylin and eosin staining of colon Swiss rolls at study termination, black arrows indicate lesions of severe inflammation/ulceration. **I** Semi-quantitative histological scores of inflammation accounting for extent of mucosal loss, hyperplasia, and erosions, *n* = 5 independent animals. Black asterisks indicate significance against DSS and grey asterisks indicate significance against *S.b. mSA*. **J** Mean relative expression of *Fn1*, *Fgb*, and *Col4a1* in distal colon of DSS mice relative to healthy controls, *n* = 4. Dotted lines represent the limit of detection and shaded regions below represent any values below the limit of detection (Panels B, C, and D). Data are shown as mean ± SD. Significance was determined using ordinary one-way ANOVA with Dunnett's (panels **B**–**E**) or Tukey's (panels **F**, **G**, **I**) multiple comparisons test, α = 0.05, \**p* < 0.05, \*\**p* < 0.01, \*\*\**p* < 0.001, \*\*\*\**p* < 0.0001. Scale bars are 100 μm.

In an acute, DSS-driven model of murine colitis, *S.b.* targeted towards fibronectin exhibited the longest gut residence time, with approximately 100-fold higher concentrations of *S.b.* in the fecal content and locally within the colon tissues than non-targeted *S.b.* or *S.b.* targeted towards fibrinogen or collagen IV. While the *S.b. FN*, *S.b.*

*FB*, and *S.b. CIV*-treated groups exhibited similar cytokine expression levels at study completion, the *S.b. FN* group displayed the highest therapeutic effects. Multiple systems, including the microbiome, metabolome, and host immune cells likely contribute to *S.b. FN's* robust therapeutic response in the acute DSS model. Further research

is needed to understand the temporal dynamics of the host immune system and cytokine expression levels in response to probiotic exposure over time. Notably, only the *S.b. FN* remained at pharmacologically relevant concentrations within the colon during the 72-h recovery period following the DSS challenge, resulting in significant improvements in mouse colon length, colonic cytokine expression profiles, and histological scores.

Between 48- and 72-h hours post-gavage, the surface binding moieties have likely been diluted or degraded, leading to elimination of *S.b.* from the gut. Importantly, for the 72-h recovery period following the DSS challenge, *S.b. FN* was maintained at concentrations high enough within the colon to elicit a pharmacologically relevant response. Mouse colon length, colonic cytokine expression profiles, and histological scores were all significantly improved in mice treated with *S.b. FN*.

In the relapsing model, we observed that within the first 24 and 48 h after dosing, the CFU load in the feces was highest in the *S.b. FN*-treated group. This observation reflects the trends seen in the acute DSS model (Fig. 4). However, at later timepoints, the *S.b. CIV*-treated group exhibited the highest fecal CFU load, consistent with data demonstrating the highest Col4a1 expression in the colon (Fig. 5J). This shift suggests changing expression levels of fibronectin to collagen IV within the colon. It also emphasizes the potentially important role that ECM remodeling plays in probiotic residence time. At the conclusion of our relapsing study, we observed a reduction in inflammatory markers (such as weight, colon length, and cytokine expression levels) in the *S.b. CIV*-treated group, indicating that prolonged probiotic exposure significantly improved recovery in mice. However, interestingly, there was no corresponding decrease in fecal CFU. One possible explanation for this discrepancy is that although the inflammation was mitigated, it did not fully return to healthy baseline levels. Hence, even partial upregulation of Col4a1 could contribute to prolonged *S.b. CIV* retention. The spatiotemporal dynamics of the ECM remodeling process within each preclinical colitis model of colitis need to be further characterized in future studies to fully elucidate the relationship between the ECM and targeted probiotic residence time.

Studies have shown that the ECM in the inflamed GI tract is dynamic in nature and its expression, deposition, and remodeling changes in response to disease progression, severity, and degree of inflammation within the lesion[40–42,44,45]. Moreover, upregulated ECM in inflammatory lesions plays a pivotal role in downstream signaling, triggering additional inflammation. This perpetuates a challenging cycle of inflammation. While the body increases ECM protein production to structurally safeguard the area and promote healing, dysregulated and excessive deposition of these proteins leads to further inflammation and an inability to heal the lesion[41,42,92]. Interventions aimed at regulating excessive ECM production may offer an alternative or complementary strategy for future therapeutic development. However, given the critical role of ECM in the healing process, achieving a fine-tuned balance will be essential.

The dynamic ECM remodeling process calls for not only patient-specific but also lesion-specific targeting of therapeutic microbes. Our platform approach has the potential to address this need. It allows for the PK/PD profiles of different ECM-targeted *S.b.* to be easily explored under differing disease severities because the targeting ligand can be rapidly swapped out. Further, patient-specific ECM targeting ligands could be identified through individual biopsy analysis, potentially enhancing probiotic retention and efficacy in a patient-specific manner.

Our study represents a tunable targeting approach to enable *S. boulardii* to bind to the extracellular matrix, resulting in a remarkable 100-fold increase in colon retention. Previous work reports rapid clearance of *S. boulardii* from the murine colon, especially without the support of antibiotic co-administration[27,28]. Here, through ECM-targeting, we elongate the probiotic gut residence time by 24–48 h without antibiotic co-administration. As we move forward, it will be

crucial to investigate how ECM-targeting of *S.b.* affects probiotic efficacy in immune-driven pre-clinical models of ulcerative colitis and Crohn's disease. Notably, the relative expression of fibronectin, fibrinogen, and collagen IV within the murine colon in the DSS models of colitis is parallel to what is found in relative expression values in colonic biopsies from patients with active colitis. While the DSS-driven models provide initial support for improved probiotic efficacy using ECM-targeted *S.b.* and key insights into in vivo probiotic pharmacokinetics, these models have limitations in how well they model human disease. Rather than rely upon chemical injury, exploration of *S.b.* PK/PD in models such as the *Il10*$^{-/-}$ knockout, T-cell transfer, and using IBD patient fecal microbiota transfer-based models will be critical in establishing the validity of this approach[5,6]. Our findings suggest that *S.b.* stimulates the immune system in an anti-inflammatory manner, but the precise mechanisms behind this effect remains unclear[7]. Thus, the therapeutic mechanisms of action of *S.b.* need to be more characterized to safely scale up this approach for human use. Several reports indicate the importance of the microbiome composition and diversity in disease progression and treatment response for inflammatory bowel diseases[93–95]. However, the role of the microbiome in IBD progression remains incompletely understood within the field. Further research is necessary to comprehensively assess microbiome changes throughout disease progression and in response to treatment, allowing a deeper understanding of treatment impact on microbiome diversity dynamics.

Notwithstanding these caveats, this work provides impactful insights for improving the therapeutic potential for live biotherapeutic products through the application of engineering and drug delivery principles. In this study, we successfully demonstrated that the targeted *S.b.* enables a significant reduction in dosing frequency, moving away from the daily reported dosing schedules[46,76,81], to an interval of once every three days. Furthermore, we show significant recovery in both acute and chronic DSS models of colitis with the targeted *S.b.*. We posit that this platform can be further improved through stable expression of ECM-specific ligands on the yeast cell surface, thereby enabling continuous targeting capabilities with fewer doses. Moreover, a genetic toolkit has been established for *S.b*[27], which allows for further genetic engineering for drug molecule secretion to establish in situ, lesion-targeted drug delivery. Collectively, this work presents a tunable method for modulating probiotic pharmacokinetics, binding site-specificity, and efficacy with the ultimate goal of advancing the development of patient-friendly and disease-specific microbiome-based therapeutics for inflammatory bowel diseases.

## Methods

Animal studies were conducted in accordance with and approved by the Institutional Animal Care and Use Committee (IACUC) of The University of North Carolina at Chapel Hill.

### Strains and media

*E. coli* DH10B were used for plasmid construction, maintenance, and propagation. *E. coli* were grown in Luria Bertani medium containing 100 µg/mL ampicillin at 37 °C with aeration. *S. boulardii* strains with the auxotrophic marker, ura3, were kindly donated by Prof. Yong-Su Jin[96]. The *S. boulardii* strain engineered to stably express mSA on the cell surface is termed *S.b. mSA*. *S. cerevisiae* strain JK9-3dα was used as a control yeast strain in probiotic mechanisms of action assays. Yeast were grown in yeast extract peptone dextrose (YPD, Difco) medium at 30 °C with aeration. Engineered *S. boulardii* were selected on synthetic dropout agar plates (SD; 20 g/L glucose, 1.71 g/L yeast nitrogen base (without amino acids), 5 g/L ammonium sulfate, and 0.74 g/L complete supplement mixture-uracil (CSM-Ura, Sunrise Science Products)). For plate attachment assays, *S. boulardii* was incubated in complete minimal media (CMM; 20 g/L glucose, 1.71 g/L yeast nitrogen base (without amino acids), 5 g/L ammonium sulfate, and 0.74 g/L complete

supplement mixture). For the growth under physiological challenges assay, yeast strains were grown in CMM adjusted to either pH 2.5 or pH 4.0 with hydrochloric acid or were supplemented with 0.3% w/v or 0.6% w/v Oxgall (Difco) dehydrated bile. For the labeling and attachment efficiency time course assay, *S. boulardii* were grown in simulated intestinal fluid (Ricca Chemical Company) supplemented with 25% v/v CMM. All yeast media was supplemented with 100 U/mL penicillin and 100 µg/mL streptomycin to prevent bacterial growth. For CFU analysis, fecal samples or tissue homogenates were serially diluted and plated on YPD agar plates supplemented with 2.5 µg/mL ampicillin, 2.5 ug/mL gentamicin, 2.5 ug/mL metronidazole, 2.5 µg/mL neomycin, and 1.25 µg/mL vancomycin.

## Cell culture

Murine bone marrow-derived dendritic cells (BMDCs) were cultured as described previously[97]. Briefly, bone marrow obtained from 8- to 12-week-old C57BL/6 mice was washed and seeded in BMDC differentiation media composed of RPMI 1640, L-glutamine, 10% FBS, and 100 U/mL penicillin, 100 µg/mL streptomycin, and mouse granulocyte-macrophage colony-stimulating factor (GM-CSF). Media was refreshed every other day for 10 days. Thereafter, BMDC media containing IL-4 at 10 ng/mL was used to stimulate dendritic cell differentiation for 4 days. After a total of 14 days, nonadherent cells were seeded in 24-well plates at $3 \times 10^5$ cells/well in GM-CSF-free media overnight prior to the immune modulation assay described below.

## Plasmid Construction and Yeast Transformation

pIU211 (Addgene Plasmid #44036) containing an AGA1 expression cassette was modified to replace the galactose-inducible promoter sequence with a pTDH3 sequence to confer constitutive expression of AGA1. The pTDH3 promoter sequence was PCR amplified from MoClo-YTK (Addgene Kit # 1000000061). pYD1-mSA (Addgene Plasmid #39865) containing a AGA2-monomeric streptavidin (mSA) expression cassette with flanking HA and FLAG tag sequences was modified to replace the galactose-inducible promoter sequence with a pTPI1 sequence to confer constitutive AGA2-mSA fusion protein expression. The pTPI1 promoter sequence was PCR amplified from a plasmid in-house[98]. The constitutive AGA2-mSA expression cassette was then PCR amplified with primers modified to include a 5′ SacI and a 3′ KasI restriction enzyme cutting sites. The amplified product and the modified pIU211 plasmid were cut with SacI and KasI and were then ligated together utilizing T4 ligase to generate pConst-mSA-Display plasmid. The plasmid was sequenced to confirm that the plasmid contains two expression cassettes: 1) constitutive AGA1 expression and 2) constitutive AGA2-mSA expression with a URA3 marker to enable stable genomic integration into ura3 auxotrophic *S. boulardii*. The integrative plasmid was linearized with the restriction enzyme, BsiWI, to direct genomic integration. Competent *S. boulardii* were prepared and transformed with linearized DNA as previously described using the LiAc/PEG method[99]. Briefly, *S.b.* were inoculated in 50 mL fresh YPD medium to reach an $OD_{600}$ of 1.6 and were then collected via centrifugation. *S.b.* were washed 3 times with 50 mL sterile water and resuspended with 1 mL sterile water and then separated into 100 µL aliquots. For transformation, 360 µL of "transformation mix" containing PEG 3350 (50% w/v), lithium acetate (1.0 M), single-stranded carrier DNA (2.0 mg/mL), and the linearized plasmid DNA (10 µg) was added to the prepared aliquot of competent *S.b.* This mixture was incubated in a water bath at 42 °C for 60 min. Transformed *S.b.* were then collected via centrifugation, resuspended in sterile water, and plated on selective SD-Ura agar plates lacking uracil. Plates were incubated for 3 days at 30 °C. For biotin-plate binding studies in Fig. 1, a control strain was generated (labeled *S.b.* in Fig. 1), wherein the mSA sequence was cut out using NheI and BamHI and replaced by a short glycine-serine linker flanked by NheI and BamHI to enable HA and FLAG tag surface expression without mSA.

## In Vitro Plate Binding Assays

Biotin Well Plate Attachment: A Pierce Biotin-Coated 96-well plate was blocked for 1 h with 1% bovine serum albumin solution in phosphate-buffered saline (PBS). The plate was washed 3x with PBS. Then, a saturated culture of *S. boulardii* strains grown in YPD overnight, was washed with PBS to remove excess media, and were diluted to the indicated $OD_{600}$ in CMM. Diluted cultures were incubated at 100 µL/well for 1 h at 30 °C.

Extracellular matrix (ECM) protein binding assays: Nunc MaxiSorp flat-bottom 96-well plates were incubated with 200 µg/mL of either fibronectin, fibrinogen, or collagen IV proteins (Abcam, ab81743, ab92791, or ab7536, respectively) in a sodium bicarbonate buffer at pH 9.4 overnight at 4 °C. Plates were then blocked for 1 h with 1% bovine serum albumin solution in PBS. Saturated cultures of *S. boulardii* strains were washed as previously described. Cells were then incubated with 100 nM of either biotin-anti-fibronectin, biotin-anti-fibrinogen, or biotin-anti-collagen IV (Abcam, ab6584, ab51416, or ab6581, respectively) for 30 min, then washed with PBS to remove unbound antibodies. The antibody-labeled cells were then resuspended in CMM and incubated at 100 µL/well at the indicated cell concentrations and the corresponding ECM protein-coated well plates. For the nonspecific (NS) biotinylated antibodies in each assay: biotin-anti-fibrinogen was used in the fibronectin and collagen IV attachment assays, and biotin-anti-fibronectin was used in the fibrinogen attachment assay. Each plate was incubated statically for 1 h at 30 °C.

Attachment Time Course Assay: A fibronectin-coated well plate was prepared as previously described. A saturated culture of *S.b. mSA* was washed with PBS and labeled with biotin-anti-fibronectin as previously described. Labeled *S. boulardii* were then incubated in supplemented simulated intestinal fluid for 48 h with orbital shaking at 30 °C. At 12-h increments, aliquots of culture were subject to either flow cytometric analysis or incubated in the fibronectin-coated well plate. At each time point, for the plate binding assay, aliquots were centrifuged and resuspended in CMM to reach a final concentration of $OD_{600} = 1$. These samples were incubated at 100 µL/well in the fibronectin-coated well plate for 1 h at 30 °C.

For all attachment studies, after incubation with *S. boulardii*, wells were washed 2x with PBS to remove unbound *S. boulardii*. Since HA is co-expressed with mSA on the cell surface, bound cells were then incubated with a 1:500 dilution of anti-HA-AlexaFluor 488 antibodies (R&D Systems, IC6875G) in PBS to label the attached cells in green. Wells were washed once more with PBS to remove unbound fluorescent antibody, then subject to fluorescent imaging analysis. Fluorescent microtiter plate images were taken from the center of each well at 10x magnification (Revolve, Echo). Each condition was tested in triplicate. Image analysis was conducted using Particle Counting in ImageJ.

## Flow cytometry

Overnight cultures of *S.b. mSA* grown in YPD were centrifuged at 3000 rpm (700 x g) for 3 min and resuspended in PBS containing 0.05% Tween 20 (PBST). Cells were diluted in PBST to reach a final $OD_{600} = 0.05$. For each condition, 1 mL of diluted cells were centrifuged at 13,000 x g for 1 min. For detection of mSA expression on the cell surface, the cell pellet was resuspended in 50 µL of 1:50 dilutions of anti-HA-Alexa Fluor 488 and anti-FLAG-Alexa Fluor 594 in PBST since HA and FLAG flank mSA and are co-expressed on the yeast surface with mSA. Cells were incubated with the antibodies for 30 min at room temperature, washed with PBST, then subject to flow cytometric analysis. For the biotinylated antibody dissociation constant ($K_D$) measurements, cells pellets were prepared as described then resuspended in 50 µL of the indicated biotinylated antibodies at concentrations ranging from 1000 nM to 6 pM in PBST. A non-biotinylated anti-fibronectin antibody (Abcam, ab2413) was used as a control. Cells were incubated at room temperature for 30 min. Cells were then washed with PBST and resuspended in 50 µL of a 1:100 dilution of anti-rabbit-Alexa Fluor 594 for

30 min on ice. Cells were then washed with PBST and subject to flow cytometric analysis. For all flow cytometry experiments, a LSRII Fortessa was used with lasers at 488 nm and 561 nm.

## In vitro probiotic mechanisms assays

Resistance to Physiological Challenges (Fig. 3B): Overnight cultures of yeast strains were diluted in CMM media containing the indicated pH and bile salt challenges to a yeast concentration of $OD_{600} = 0.6$. Yeast was cultured at 30 °C with aeration for 4 h. Cultures were then serially diluted and plated onto YPD agar plates and incubated at 30 °C for 2–3 days for viable CFU enumeration.

## Short chain fatty acid secretion

Stimulation of short-chain fatty acid production (Fig. 3C–E): Overnight cultures of *S.c.*, *S.b.*, or *S.b. mSA* were diluted into fresh CMM to reach a final OD600 = 0.3. *S.b. mSA* cultures were centrifuged and incubated with 100 nM of biotin-anti-fibronectin, biotin-anti-fibrinogen, or biotin-anti-collagen IV (Abcam, ab6584, ab51416, or ab6581, respectively) for 30 min, then washed and resuspended with CMM. Cultures were incubated at 37 °C for 18 h in aerobic conditions. Cultures were then centrifuged, and aliquots of media supernatant were collected 6, 12, and 18 h post-CMM inoculation for LC-MS analysis of SCFAs. A volume of 100 µl cell media was used for derivatization with 10 µl 2.4 M aniline and 10 ul 1.2 M N-(3-Dimethylaminopropyl)-N0-ethylcarbodiimide hydrochloride (EDC) and placed on ice for 2 hr[100]. Acetate, propionate, and butyrate were quantified on an AB SCIEX (Foster City, CA) QTRAP 6500 system consisting of a SHIMADZU Nexera ultra-high-performance liquid chromatography system coupled with a hybrid triple quadrupole and ion trap mass spectrometer. Analyst® 1.6 software was used for system control and data acquisition. Data processing and quantitation were performed with the MultiQuant 3.0 software. A straight line was fitted to the linear portion of secreted SCFA amount before saturation was reached (0 to 18 h for acetate, 0 to 12 h for butyrate and propionate). The slopes of the linear regression were used to assess the secretion rates of SCFA.

## Immune modulation assay

Overnight yeast cultures were centrifuged, washed with DPBS, and resuspended in fresh BMDC media to reach a final $OD_{600} = 0.02$. Cell culture media from a 24-well plate containing differentiated BMDCs was removed and replaced with 500 µL of yeast suspension per well ($1 \times 10^5$ yeast cells/well). Control wells were loaded with PBS or lipopolysaccharide (1 µg/mL). Yeast and controls were incubated with the BMDCs at 37 °C for 16 h. Culture supernatant was collected for enzyme-linked immunosorbent assay (ELISA) analysis of BMDC-secreted IL-10. An IL-10 mouse ELISA kit (Invitrogen) was used to assay the supernatants and the optical density at 450 nm was read using a SpectraMax iD3 96-well plate reader.

## IL-8 suppression assay

Experiments were conducted 24 h after cells were seeded on six-well plates at a concentration of $1 \times 10^4$ cells/well. *S.b.* from overnight cultures were centrifuged, washed 3x with sterile PBS, resuspended in RPMI, and added at a concentration of $5 \times 10^5$ cells/well. *S.b.* was incubated with HT-29 cells for 18 h. Cells were then stimulated with 20 ng/ml of tumor necrosis factor-alpha (TNF-a) (Sigma, GF314) for 6 h. Supernatants were collected for enzyme-linked immunosorbent assay (ELISA) analysis. A human IL-8 ELISA kit was used to assay the supernatants and the optical density at 450 nm was read using a SpectraMax iD3 96-well plate reader.

## Animal studies

Six- to eight-week-old female C57BL/6 J mice housed in groups of 5 were used for all in vivo studies. Mice were purchased from The Jackson Laboratory. Mice were fed the standard chow, Select Rodent Diet 50 IF/6 F Auto Diet (LabDiets) throughout all studies. Mice were separated into 2–3 mice per cage and acclimated for at least 72 h prior to study initiation. For oral administration, yeast strains were cultured overnight in YPD media at 30 °C (with shaking at 250 rpm). Yeast pellets were collected via centrifugation at 3000 rpm (1200 x g) for 3 min, washed twice with sterile saline, and resuspended in sterile saline. Aliquots of the resuspended yeast slurries were kept for serial dilution plating and CFU quantification. For all oral doses of yeast, mice were administered $10^9$ CFU in 150 µL sterile saline. Mouse weight was monitored daily.

Acute DSS Colitis Model: Mice received 2% colitis grade dextran sulfate sodium (DSS) salt (MP Biomedicals, Cas. No. 9011-18-1) *ad libitum* in the drinking water for 5 days. Fresh DSS solution was prepared, and filter sterilized every other day. On day 5, mice were placed back into normal drinking water and received a dose of $10^9$ CFU of yeast. On day 8, mice were euthanized, and tissues were harvested for downstream analysis.

Chronic DSS Colitis Model: Mice received 2% DSS in the drinking water as described previously for 5 days. This was followed by a 3-day "recovery" period where mice were placed back on the normal drinking water. The 5-day "challenge", 3-day "recovery" cycle was repeated twice more. Beginning on day 9, mice received $10^9$ CFU of yeast every 3 days for the remainder of the study. On day 24, mice were euthanized, and tissues were harvested.

## Quantitative culture for fecal and tissue colony forming unit analysis

At indicated time points, each mouse was individually placed in a sterilized cage and approximately 2–4 fecal pellets were collected from each mouse. Pellets were placed in pre-weighed homogenization tubes containing 1.4 mm ceramic beads. Tubes were weighed again to calculate fecal weight, and 500 µL sterile PBS was added to each tube. Feces were then homogenized with a FastPrep-24 homogenizer (MP Biomedicals) for 20 s at 4.0 M/s. Tissue samples were placed in homogenization tubes with 200 µL sterile PBS and 2.4 mm ceramic beads, then homogenized for 45 s at 6.5 M/s. Homogenates were serially diluted 10-fold in sterile PBS, and dilutions were plated on YPD agar plates containing ampicillin, gentamicin, metronidazole, neomycin, and vancomycin[51]. CFUs were enumerated after incubation for 2–3 days at 30 °C. CFU enumeration in the fecal and tissue samples was normalized to either the fecal or tissue weight of the sample. CFU counting and limit of detection standards were used as previously described[98].

## mRNA extraction and qPCR

For relative expression analysis of colonic cytokines and genes encoding ECM proteins, samples from the distal colon tissue from each mouse were collected in homogenizer tubes containing 2.4 mm ceramic beads and 200 uL sterile PBS. Tissues were then homogenized with a FastPrep-24 homogenizer (MP Biomedicals) for 60 s at 6.5 M/s. To each tube, 800 uL of Qiazol (Qiagen) reagent was added and incubated for 5 min. Thereafter, a standard phenol/chloroform extraction protocol was followed to recover colonic mRNA. Complementary DNA (cDNA) was generated using iScript cDNA Synthesis Kit (Biorad) following the manufacturer's standard protocol. For all qPCR reactions, iTaq Universal SYBR Green Supermix (Biorad) was used following the manufacturer's standard protocol. All qPCR reactions were measured using a QuantStudio 6 Flex Real-Time PCR System (Applied Biosystems). All samples were analyzed in duplicate. Technical duplicates were averaged to calculate a Ct value for each sample. Ct values were then normalized to the housekeeping gene *ActB* within each sample, and fold changes were calculated by the delta-delta Ct method.

## Histological evaluation

At study endpoints, mice were euthanized, and gastrointestinal tissues were collected. Colon tissues were photographed, and colon lengths were measured from the rectum to the cecum using ImageJ. Colon tissues were then oriented in a standard Swiss Roll formation and fixed in 10% neutral buffered formalin for 24 h. Tissues were then processed, and embedded in paraffin wax, and 5-um sections were obtained following standard FFPE procedures. For colonic biodistribution following acute DSS colitis, colon tissues were snap-frozen into OCT, sectioned at 5–10 um thickness using a cryostat, fixed in Carnoy's solution, then stained with anti-fibronectin (Proteintech, clone 1G10F9) and anti-Saccharomyces antibodies (BioRad, 8203-0050) at a 1:100 dilution, followed by fluorescent secondary antibodies (Invitrogen, A-11032 and A-21245) at a 1:1000 dilution, and finally DAPI/Fluoromount. Fluorescently stained tissues were imaged using an Echo Revolve microscope using Texas-Red, Cy5, and DAPI channels. Histological inflammation scores were based on the severity of mucosal loss, mucosal epithelial hyperplasia, degree of inflammation, and extent of pathology, as previously described[84,85]. Histological inflammation scoring was performed blindly by a licensed, board-certified veterinary pathologist.

## Whole-genome transcriptional analysis

(Figure 1F) The following datasets were accessed via the Gene Expression Omnibus (GEO) and utilized for a pooled expression analysis: GSE13367, GSE9452, GSE38713, GSE47908, GSE73661, GSE114527, GSE87466. The interactive web tool, GEO2R, was used to define control and colitis samples and to extract expression values. Expression values for fibronectin 1 (GeneID 2335), collagen type IV alpha 1 chain (GeneID 1282), and fibrinogen beta chain (GeneID 2244) were used for downstream analysis. Fold-change was first calculated for each sample and each ECM protein within individual datasets. Fold-change values were then pooled and plotted for a comprehensive expression analysis from all the datasets.

## Data and statistical analysis

The number (n) of replicates/animals used per group is described in each figure legend and represents biological replicates. All well-plate binding studies were performed 3 times independently. All dilutions for CFU analysis were plated in technical duplicates. Samples for ELISA were run in technical duplicates. All qPCR samples were run in technical duplicates. All statistics and data distribution analyses were performed with Prism (GraphPad). The unpaired two-tailed Student's t-test was used to compare differences between the two groups. Ordinary one-way ANOVA with multiple comparisons with Šídák's, Tukey's, or Dunnett's multiple comparisons tests (as indicated in figure legends) was used to evaluate experiments containing more than two groups. The upper threshold for statistical significance for all experiments was set at $p < 0.05$.

## Generative AI and AI-assisted technologies in the writing process

During the preparation of some parts of this manuscript, the author used Microsoft Copilot to improve the readability of the manuscript. After using this tool, the authors reviewed and edited the content as needed and take full responsibility for the content of the publication.

## Reporting summary

Further information on research design is available in the Nature Portfolio Reporting Summary linked to this article.

## Data availability

The transcriptional expressions of Fn1, Col4a1, and Fgb from healthy and UC human patients were analyzed from the published GEO datasets (http://www.ncbi.nlm.nih.gov/geo/) obtained from GSE13367, GSE9452, GSE38713, GSE47908, GSE73661, GSE114527, and GSE87466. The data underlying Fig. 1d–f, Fig. 2b, e, f, Fig. 3b–f, Fig. 4b–f, h, i, Fig. 5b–g, i, j, as well as Supplementary Figs. 2–5 are in the associated Source Data file. Source data are provided with this paper.

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

## Acknowledgements

We acknowledge funding from the UNC Chapel Hill Center for Gastro-intestinal Biology and Disease pilot/feasibility grant (5124948, through the NIH P30 DK034987) and UNC start-up funds. JN was in part funded by the NIH R01CA241679. Parts of Fig. 1–6 were created using Bior-ender.com. We acknowledge the following individuals for their con-tributions: Aabir Sanyal for assistance in sample organization for in vivo studies, and Sean Simpson, PhD in the Ainslie lab at UNC Chapel Hill for obtaining and culturing murine bone marrow-derived dendritic cells with funds from the NC Biotech Center (2021-FLG-3822). We acknowledge Yong Su Jin, PhD, professor of Food Microbiology at the University of Illinois at Urbana-Champaign for kindly donating *S. bou-lardii* strains with auxotrophic markers. We acknowledge assistance from the following core facilities at UNC Chapel Hill: Pathology Services Core (Rani Sellers, DVM, PhD, DACVP) for histological evaluation and scoring of colon tissues supported in part by an NCI Center Core Sup-port Grant (P30CA016086); and the UNC Flow Cytometry Core Facility for cytometer access. We thank Dr. Matthew Bolton from the University of Virginia and Dr. Chelsea Anderson (UNC) for their expertise in statis-tical analyses. Services in support of the research project were provided by the VCU Massey Cancer Center Lipidomics and Metabolomics Shared Resource, supported, in part, with funding from NIH-NCI Cancer Center Support Grant P30 CA016059. YW acknowledges financial support through the T32CA196589 fellowship. AH acknowledges financial sup-port through the T32GM135095.

## Author contributions

Research idea, experimental study concept and design, M.H., J.N., J.A. and A.A. Acquisition of data, M.H., M.G. and A.H. Statistical analysis, M.H., Y.W. and J.N. Drafting and writing of the manuscript, M.H., J.N. and J.A. Manuscript editing, revisions, and review M.H, J.N., J.A., and Y.W. Study supervision, J.N. and J.A. Funding, J.N. J.A., and A.A.

## Competing interests

J.N., M.H., and A.H. are inventors of the patent applications of the tar-geted *S. boulardii* technology evaluated in this paper. These relationships have been disclosed to and are under management by UNC Chapel Hill. The remaining authors declare no competing interests.
