## [Peer Review File · Nature Communications]

REVIEWER COMMENTS

Reviewer #1 (Remarks to the Author):

The use of live biotherapeutics in the treatment of many conditions including inflammatory bowel disease is emerging as a potential novel therapeutic opportunity. A key challenge in these applications is overcoming natural community stability and achieving long-term stable colonisation of therapeutic microbes. In this context, *Saccharomyces boulardii* is currently employed as a probiotic option with clinical trials showing positive results in the treatment of inflammatory bowel disease. In this study the authors have modified *S. boulardii* to improve binding to inflammation associated markers within the GI tract. Combining in vitro and in vivo experimental models the authors have achieved prolonged colonisation and demonstrated therapeutic benefit using the DSS mouse model of colitis. The authors should be commended for the substantial body of work and the thoroughness in testing multiple ECM targeted strains. This stated, while the use of mSA, FN, FB and CIV strains all have merit, the inconsistent use of strains through different assays, further justification of a number of key assays, and more detailed analysis of the microbiome would be beneficial.

Major Comments:

1. The authors provide data suggesting *S.b.* and *S.b.* mSA exhibit similar fold induction for commonly measured short chain fatty acids (Acetate, Propionate, Butyrate); however, these measurements were taken only at 18hrs in co-culture. Could the authors provide further justification for selection of species and time point selection? While it will obviously never be possible to demonstrate a lack of change conclusively, more detailed analysis across multiple time points and potentially in co-culture with human GI bacteria rather than mouse could be consider to provide key support to the conclusion presented.
2. To characterise the conserved host immune response the authors examine IL10 response in bone marrow derived murine dendritic cells for *S.b.* mSA and *S.b.* FN. Could the authors provide further justification for this assay as a demonstration of the conserved host response given the dominant mechanism of action reported for *S.b.* probiotics action is thought to be through inhibition of NFKB, activation of PPAR- γ and interaction with MAP kinases (ERK1, 2, p38, etc.). Also, what evidence exists to ensure *S.b.* FN is representative of *S.b.* FB and *S.b.* CIV.
3. In both the DSS model (Figure 5) and the amplicon sequencing experiments (Figure 6) comparison appears to be made to the DSS condition (without *S.b.*) as opposed to the *S.b.* mSA strain. Further justification for this analysis decision is necessary given *S.b.* alone is known to exhibit beneficial impact. Given the key questions were focused on the improved efficacy of binding and subsequent

improvement (if any) in therapeutic efficacy as a result of this prolonged colonisation, a comparison to the DSS treatment with S.b. mSA in addition to the DSS treatment alone would appear justified.

4. While S.b. FN demonstrates higher faecal CFU counts at 48hrs and 72hrs in the acute DSS model suggesting FN supports improved colonisation and corresponding improvements in body weight post gavage, it is notable the gene expression signatures (Figure 4F) appear to be more strongly correlated between FB, CIV and FN. Could the authors provide further details to explain these data? Further details on experimental design and timing would also aid interpretation of these host responses.

5. Can the authors provide further explanation on the observation that only FN strains exhibit improved colonisation by fecal CFU? Given the key impact of the modified strains would be expected to be correlation between GI inflammation and stable colonisation, would one not expect to observe improved/prolonged colonisation with all the ECM modified strains. Does this reflect incongruence between faecal shedding and load at site of disease or does it indicate only S.b. FN target was suitably presented in the host to achieve sustained colonisation?

6. In Figure 4J, while FN exhibits localisation, presentation (in primary Figure or supplementary) of FB and CIV data would be beneficial to assess both if faecal CFU load correlates with colonic load and variation between FB, CIV and FN and related association with phenotypic outcomes. Image quantification could also be beneficial in this context.

7. The comparison of the acute and relapsing model provides significant insights into the mechanisms of action. This stated, more detailed explanation and rationale for the first S.b. gavage being undertaken at day 9 in the relapsing model as opposed to day 5 in the acute model could aid interpretation of this comparison.

8. In all the faecal CFU counts obtained in the relapsing DSS mode the 48hrs and 72hrs post dose CFU counts vary from below detection to 10^5 CFU/g. Has this resulted from combining datasets across the time course (i.e. 48hrs Post-dose represents Day 11, Day 14, Day 17, etc.) or does this data represent a single time point with a clear dichotomy in colonisation success? The analysis and/or visualisation of this data should be modified to improve clarity and represent the data for each time point considered. Similarly, including the CFU curves, probably as supplementary, from which the data for 5C was calculated would provide a clearer indication of stability of colonisation over time.

9. In the relapsing DSS model, given the decreased inflammation (and therefore CIV binding opportunity) over the 25 days - was a reduction in CFU load observed as the inflammatory state was resolved?

10. Examination of the response within the gut microbiome has the potential to provide important understandings into the capacity of S.b. CIV to modulate disease outcome. Unfortunately, the amplicon sequencing analysis has only provided minimal insight. Could this analysis be further expanded to explore key microbe dynamics over time. Variation between mice should also be clearly presented. Importantly, the relevant comparison would be S. b. mSA compared to S.b. CIV, ideally with S.b. FN and S.b. FB also included in the analysis.

Minor points:

1. In the abstract the authors state these studies demonstrate the potential for "patient-friendly" treatments for IBD, it is unclear how this study provides this outcome.

2. Line 302: Grammatical error: "as S.b. FN did not persist within the healthy colon as long in a healthy mouse"

2. Line 658: Grammatical error: "for 2 minutes. System then heated at 40 degrees"

Reviewer #2 (Remarks to the Author):

The manuscript by Heavey et al. describes the development of a targeted probiotic yeast platform that attaches to extracellular matrix (ECM) proteins present in inflammatory lesions of the murine GI tract. Using a biotin-streptavidin capture the authors attach antibodies specific for 3 ECM proteins. In murine models of ulcerative colitis, the engineered yeast platform binding to fibronectin demonstrated significantly longer residence time in the gut and increased abundance in the colon compared to fibrinogen and collagen IV binding strains as well as the strain not conjugated with antibodies. The fibronectin and collagen IV binding strains demonstrate benefits in short and long term DSS model. The results include increased colon length, improved colonic cytokine expression profiles, reduced histological inflammation scores, and enhanced microbiome diversity indices. The study provides an interesting approach to improving residence time and lays the groundwork for future work on improved engraftment and target specific treatment.

General comments:

- To avoid confusion and ensure accurate interpretation of the data presented, the authors should maintain consistency in the statistical methods used throughout the manuscript. This is especially important in Figure 4 and 5, where two different multi comparison tests are used inconsistently. Additionally, the legend should state the reference/control when a Dunnett's test is used.
- The authors should provide more information on the potential involvement of ECM proteins in regulating inflammation, as well as the potential anti-inflammatory effects of dosing antibodies alone. This would justify the lack of an antibodies alone control group.
- It would be preferable for the authors to conduct a follow-up test to confirm that the S.b. mSA coating does not permanently engraft in the inflammatory induced mice model. This would help ensure the safety of the method and provide more insights into the dynamics of the colonization profile.

Major comments:

Line 52. Engineered probiotics and LBPs are not the same thing. In addition, an LBP doesn't necessarily need to be a probiotic microbe. The term "engineered live biotherapeutic products (eLBPs)" is a commonly used term that could be adopted for clarity. The authors should also use consistent terminology throughout the manuscript in response to "engineered therapeutic microbes" on line 65.

Line 113. The authors should acknowledge that removing fungal infections with anti-fungal agents is not always a quick and easy solution, and that some anti-fungal agents can also impact gut bacteria, as seen in mice (doi: 10.1128/AAC.02552-20). This statement should be modified to reduce any confusion among readers.

Line 115-116: The sentence feels a bit redundant to the previous paragraph.

Line 119-121. The authors should clarify that the statistical comparisons in the figures have only been compared to DSS-only. Likewise in similar cases where the antibody coated S.b. mSA is claimed to improve the state of the disease.

Line 313-314. The authors should use the correct multi comparison test to confirm that there is no significant difference between S.b. FN and non-DSS healthy mice. As now a Dunnett's test is used according to Figure 4E.

Line 336. The authors should also confirm whether Figure 4F was corrected for multiple comparisons of genes.

Line 358. The authors should include a statistical test for Figure 4I.

Line 366-367. The authors should consider providing a reference for the cycle of 5 days DSS and 3 days recovery. Or elaborate further on this specific dosing regimen.

Line 367. The magnitude change is closer to 1000-fold than 100-fold in Figure 5D.

Line 385. The authors should clarify whether the effect observed is significant or not. Similar described in the earlier paragraph for Figure 4.

Line 387-388. The authors should consider providing more quantitative information, similar to the previous section for Figure 4.

Line 418. The authors should specify the statistical test used for Figure 5G and Figure 5J.

Line 437. The authors should clarify whether the observed effect is due to Sb or prolonged Sb colonisation. The authors should also elaborate why Sb control was not included.

Line 462-465. The authors should experimentally show that coated S.b. mSA is cleared out to strengthen the safety of this binding method.

Minor comments:

Line 45. The authors could consider including more quantitative data on the prolonged yeast colonisation in order to provide a clearer understanding of the extent of the effect.

Line 75. The authors abbreviation of "Saccharomyces boulardii (S.b)" should be "(S.b.)" to be consistent with later abbreviation style

Line 104. Saccharomyces boulardii has already been abbreviated and therefore not necessary to do it again

Line 159. For better figure aesthetic authors should consider using consistent colour coding for the linearised plasmid in A and B; and should also state the terminator used.

Line 159. The authors should consider providing the sequence used in the supplementary would improve knowledge sharing and reproducibility among the scientific communities.

Line 199. "S.b." should be corrected to "S.b. mSA."

Line 214. The authors should include a legend for Figure 2B, similar to Figure 3B.

Line 272. The statistical comparison lines should be aligned to improve readability.

Line 336. The authors should write "Log10" instead of "log," and should clearly indicate what the dotted and shadow areas represent in the figure and text legend.

Line 530, 607, 624. The authors should use consistent terminology, such as "S.b. mSA" instead of "S.b. -mSA."

Point-by-Point Response

We thank the reviewers for their insightful comments and suggestions to improve this manuscript.

We have performed additional experiments and significantly revised the manuscript based on the feedback. Please find our detailed responses below (references are provided at the end of this document). All major revisions are highlighted in **blue** in the revised manuscript and the supplemental information.

REVIEWER COMMENTS

Reviewer #1 (Remarks to the Author):

The use of live biotherapeutics in the treatment of many conditions including inflammatory bowel disease is emerging as a potential novel therapeutic opportunity. A key challenge in these applications is overcoming natural community stability and achieving long-term stable colonisation of therapeutic microbes. In this context, *Saccharomyces boulardii* is currently employed as a probiotic option with clinical trials showing positive results in the treatment of inflammatory bowel disease. In this study the authors have modified *S. boulardii* to improve binding to inflammation associated markers within the GI tract. Combining in vitro and in vivo experimental models the authors have achieved prolonged colonization and demonstrated therapeutic benefit using the DSS mouse model of colitis. The authors should be commended for the substantial body of work and the thoroughness in testing multiple ECM targeted strains. This stated, while the use of mSA, FN, FB and CIV strains all have merit, the inconsistent use of strains through different assays, further justification of a number of key assays, and more detailed analysis of the microbiome would be beneficial.

Major Comments:

1. The authors provide data suggesting *S.b.* and *S.b.* mSA exhibit similar fold induction for commonly measured short chain fatty acids (Acetate, Propionate, Butyrate); however, these measurements were taken only at 18hrs in co-culture. Could the authors provide further justification for selection of species and time point selection? While it will obviously never be possible to demonstrate a lack of change conclusively, more detailed analysis across multiple time points and potentially in co-culture with human GI bacteria rather than mouse could be consider to provide key support to the conclusion presented.

As per the reviewer's suggestion, we have incorporated additional time points (6 hours, 12 hours, and 18 hours) and included additional groups (*S.b.*, *S.b.* mSA, *S.b.* FN, *S.b.* FB, and *S.b.* CIV) in our study. The purpose was to evaluate whether genetic modification and/or the presence of antibodies impact the secretion of commonly measured short-chain fatty acids (acetate, propionate, and butyrate) from *S.b.*. Notably, *S.b.* is capable of secreting short chain fatty acids (SCFA) itself. To directly address the central question, we opted for direct quantification of SCFA secretion from *S.b.* rather than pursuing additional co-culture experiments with GI bacteria.

Figure 3 C, D illustrates that all groups exhibit comparable secretion levels of SCFA with either no statistically significant differences or minor differences (less than 1.5-fold). Overall, our findings suggest that genetic modifications and the presence of antibodies had either no effect or only minimal effects on short-chain fatty acid secretion.

Please see new Figure 3 C, D below as well as additional revisions to the main text.

Figure 3C, D. Engineered *S. boulardii* retain probiotic mechanisms of action *in vitro*. (C) Secretion of acetate, propionate, and butyrate from cultures of either *S.b.*, *S.b. mSA*, *S.b. FN*, *S.b. FB*, and *S.b. CIV*, or blank media, $n = 4$. Solid line: simple regression of the linear portion before saturation (0 to 18h for acetate, 0 to 12h for butyrate and propionate). (D) Secretion rate of acetate, propionate, and butyrate from cultures of either *S.b.*, *S.b. mSA*, *S.b. FN*, *S.b. FB*, and *S.b. CIV*, or blank media calculated from linear regression in (C), $n = 4$. Significance was determined using ordinary one-way ANOVA with Tukey's post-test, $\alpha = 0.05$, **** $p < 0.0001$.

We have revised the main manuscript as follows:

Lines 255-265: Short-chain fatty acids are molecules produced by probiotic *S. boulardii* which are known to have potent anti-inflammatory effects on the host¹. To evaluate whether genetic modification and/or the presence of antibodies impact the secretion of commonly measured short-chain fatty acids, we measured the concentration of acetate, propionate, and butyrate in the supernatant of *S. boulardii* as a function of time 6h, 12h, and 18h. **Figure 3C, D** illustrates that all groups exhibit comparable secretion levels of propionate, with a minor difference (1.1-fold) between the *S.b. FN* and unmodified *S.b.*. Regarding butyrate expression, we observed slightly enhanced secretion compared to unmodified *S.b.*, however, the enhancement in secretion rate was minimal and smaller than 1.5-fold. Similarly, the difference we observed in acetate expression between *S.b.* and *S.b. mSA* was minimal (<1.2). Overall, our findings suggest that genetic modifications and the presence of antibodies had either no effect or only minimal effects on short-chain fatty acid secretion (**Fig. 3C and D**).

2. To characterise the conserved host immune response the authors examine IL10 response in bone marrow derived murine dendritic cells for *S.b. mSA* and *S.b. FN*. Could the authors provide further justification for this assay as a demonstration of the conserved host response given the dominant mechanism of action reported for *S.b.* probiotics action is thought to be through inhibition of NF κ B, activation of PPAR- γ and interaction with MAP kinases (ERK1, 2, p38, etc.). Also, what evidence exists to ensure *S.b. FN* is representative of *S.b. FB* and *S.b. CIV*.

Additional control groups have been assayed and included in Figure 3. The reviewer brings a pertinent point in that one of the widely reported mechanisms of action of *S. boulardii* is inhibition of the NF κ B inflammatory pathway. Thus, we have performed an additional assay to measure IL-8 cytokine levels in an HT-29 co-culture assay. It has been reported that co-culture of *S. boulardii* with HT-29 intestinal cells inhibits the production of IL-8 through the NF κ B activation pathway^{2,3}. Our results reflect those found in previous reports and indicate that modification with mSA or attachment of biotinylated antibodies on the cell surface does not inhibit this mechanism of action.

We recognize certain limitations in our study. Specifically, the comprehensive mechanisms of action for *S. boulardii* have not been conclusively described in literature. The data shown here constitutes a panel of representative assays to address key mechanisms of action. The primary objective was to assess the effects of the engineered components on fundamental mechanisms of *S. boulardii*. Our data indicate that modification with mSA or attachment of biotinylated antibodies on the cell surface does not alter key intrinsic properties of *S. boulardii*. This discussion has been added to the main text as follows:

Lines 269-274: Measurement of IL-10 production in BMDCs was chosen as an additional assay because several reports indicate that co-culture of BMDCs with *S. boulardii* stimulates the production of IL-10^{4,5}. IL-10 plays a crucial role as an anti-inflammatory cytokine in the progression of IBD. This significance has been observed not only in pre-clinical models but also in human disease progression. IL-10 acts as a natural regulator, dampening the acute inflammatory processes associated with IBD⁶⁻⁸.

We have included further explanation of this in the results sections.

Lines 281-295: Inhibition of the proinflammatory NF κ B pathway represents another mechanism in which *S. boulardii* exerts an anti-inflammatory effect in the host^{2,9-12}. It has been reported that co-culture of *S. boulardii* with TNF α -stimulated HT-29 intestinal epithelial cells inhibits the production of IL-8 through the NF κ B pathway^{2,3}. To evaluate this mechanism, we performed a co-culture assay of HT-29 cells stimulated with TNF α and the various *S.b.* strains. We measured the concentrations of IL-8 in the co-culture supernatants, a cytokine regulated by the proinflammatory NF κ B pathway¹³⁻¹⁵. These results show that co-culturing the stimulated intestinal epithelial cells with any of the *S.b.* strains resulted in decreased concentrations of IL-8 in the supernatants as compared to monocultures of stimulated intestinal cells or co-cultures with the non-probiotic yeast strain, *S.c.* (**Fig. 3F**). These results represent another mechanism of action of the probiotic *S.b.* and highlight that the genetic or biochemical changes to *S.b.* in these studies do not inhibit this mechanism of action. We acknowledge that the current literature demonstrates a wide variety of mechanisms in which *S. boulardii* exerts an anti-inflammatory effect^{16,17}. Here, we perform a panel of representative assays to assess key mechanisms of *S.b.* with the primary purpose of demonstrating that our genetic and biochemical modifications do not alter these mechanisms in *S.b.*

Fig. 3 (E) Concentration of murine interleukin-10 (IL-10) in the cell culture supernatant following an 18 h co-incubation of murine bone marrow-derived dendritic cells with controls and engineered yeast, $n = 3$. **(F)** IL-8 concentrations in stimulated HT-29 epithelial co-cultured with probiotic yeast strains. Data are shown as mean \pm SD. Significance was determined using ordinary one-way ANOVA with Tukey's for panels B, D, E, and F. For panels B, E, and F: black asterisks indicate significance against PBS and grey asterisks indicate significance against S.c., $\alpha = 0.05$, * $p < 0.05$, ** $p < 0.01$, *** $p < 0.001$, **** $p < 0.0001$.

The experiment details are included in the Methods section. **Lines 722-729:**

IL-8 Suppression Assay: Experiments were conducted 24 hours after cells were seeded on six-well plates at a concentration of 1×10^4 cells/well. S.b. from overnight cultures were centrifuged, washed 3x with sterile PBS, resuspended in RPMI, and were added at a concentration of 5×10^5 cells/well. S.b. was incubated with HT-29 cells for 18 hours. Cells were then stimulated with 20 ng/ml of tumor necrosis factor-alpha (TNFa) (Sigma, GF314) for 6 hours. Supernatants were collected for enzyme linked immunosorbent assay (ELISA) analysis. A human IL-8 ELISA kit was used to assay the supernatants and the optical density at 450 nm was read using a SpectraMax iD3 96-well plate reader.

3. In both the DSS model (Figure 5) and the amplicon sequencing experiments (Figure 6) comparison appears to be made to the DSS condition (without S.b.) as opposed to the S.b. mSA strain. Further justification for this analysis decision is necessary given S.b. alone is known to exhibit beneficial impact. Given the key questions were focused on the improved efficacy of binding and subsequent improvement (if any) in therapeutic efficacy as a result of this prolonged colonisation, a comparison to the DSS treatment with S.b. mSA in addition to the DSS treatment alone would appear justified.

We agree with this and have added further statistical analyses to compare groups not only to the DSS-alone group but also to the S.b. mSA treated group. Black stars indicate comparisons to the DSS group and grey stars indicate comparisons to the S.b. mSA treated group. These details have been included in all relevant figure captions.

We also included new data in the Supplementary Figure 4 where we performed a head-to-head comparative therapeutic efficacy study between S.b. mSA to unmodified S.b.. The study revealed no statistically significant differences between the two groups. Therefore, taking into account the results from the in vivo therapeutic studies, we have decided to proceed with S.b. mSA as the control.

Supplementary Figure 4. *Sb* and *Sb mSA* show similar effects with no statistical differences in acute DSS colitis. ECM-targeting antibodies alone exert no significant effect. (A) Mean percent body weight of mice over the course of the study. $n=5$, significance indicates comparisons to average weights of DSS-only mice at each timepoint. Arrow indicates day of DSS removal and yeast dosing. **(B)** Semi-quantitative histological scores of inflammation accounting for extent of mucosal loss, hyperplasia, and erosions, $n = 5$. **(C)** Colon lengths following administration of engineered *S. boulardii* in acute DSS colitis model. **(D)** Representative images of hematoxylin and eosin staining of colon Swiss rolls. **(E)** Colon lengths following administration ECM-targeting antibodies in acute DSS colitis model. The acute DSS model was established as described under the Methods section. When switched back to normal drinking water, mice were orally gavaged with 30 μg biotin-anti-fibronectin (Anti-FN), biotin-anti-fibrinogen (Anti-FB), or biotin-anti-collagen IV (Anti-CIV) (Abcam, ab6584, ab51416, or ab6581) antibodies resuspended in 150 μL sterile PBS. The 30 μg antibody dose was chosen as this is the equivalent dose of antibody after incubation with *S.b. mSA* for targeted *S.b.* administration in Figures 4 and 5. Data are represented as mean \pm SD. Significance was assessed using ordinary one-way ANOVA with Tukey's multiple comparisons test $\alpha = ** p < 0.01$.

4. While *S.b. FN* demonstrates higher faecal CFU counts at 48hrs and 72hrs in the acute DSS model suggesting FN supports improved colonisation and corresponding improvements in body weight post gavage, it is notable the gene expression signatures (Figure 4F) appear to be more strongly correlated between FB, CIV and FN. Could the authors provide further details to explain these data? Further details on experimental design and timing would also aid interpretation of these host responses.

We thank the reviewer for bringing up this insightful analysis. We note significant changes in gene expression levels in the *S.b. FB* and *S.b. CIV* treated groups in addition to the *S.b. FN* treated group. Further details on the timing of sample collection for the expression experiment (**Figure 4F**) have been added to the main text. Additional statistical tests have also been performed to compare groups to the *S.b. mSA* group. A further explanation of this data and our hypotheses have been included in the main text.

See Lines 492-499:

While the FN, SB, and CIV groups exhibited similar cytokine expression levels at study completion, only the FN group displayed the highest therapeutic effects. Multiple systems, including the microbiome, metabolome, and host immune cells likely contribute to *S.b. FN*'s robust therapeutic response in the acute DSS model. Further research is needed to understand the temporal dynamics of the host immune system and cytokine expression levels in response to probiotic exposure over time. Notably, only the *S.b. FN* remained at pharmacologically relevant concentrations within the colon during the 72-hour recovery period following the DSS challenge, resulting in significant improvements in mouse colon length, colonic cytokine expression profiles, and histological scores.

5. Can the authors provide further explanation on the observation that only FN strains exhibit improved colonisation by fecal CFU? Given the key impact of the modified strains would be expected to be correlation between GI inflammation and stable colonisation, would one not expect to observe improved/prolonged colonisation with all the ECM modified strains. Does this reflect incongruence between faecal shedding and load at site of disease or does it indicate only *S.b. FN* target was suitably presented in the host to achieve sustained colonisation?

The reviewer brings up a great point regarding fecal shedding and loading at the disease site in comparison to ECM target expression in the host. A further explanation has been included in the main text.

Lines 505-529:

In the relapsing model, we observed that within the first 24 and 48 hours after dosing, the CFU load in the feces was highest in the *S.b. FN* treated group. This observation reflects the trends seen in the acute DSS model (Supp. Figure 5A). However, at later timepoints, the *S.b. CIV* treated group exhibited the highest fecal CFU load, consistent with data demonstrating highest Col4a1 expression in the colon (Fig 5J). This shift suggests changing expression levels of fibronectin to collagen IV within the colon. It also emphasizes the potentially important role that ECM remodeling plays in probiotic residence time. At the conclusion of our relapsing study, we observed a reduction in inflammatory markers (such as weight, colon length, and cytokine expression levels) in the *S.b. CIV* treated group, indicating that prolonged probiotic exposure significantly improved recovery in mice. However, interestingly, there was no corresponding decrease in fecal colony-forming units (CFU). One possible explanation for this discrepancy is that although the inflammation was mitigated, it did not fully return to healthy baseline levels. Hence, even partial upregulation of Col4a1 could contribute to prolonged *S.b. CIV* retention. The spatiotemporal dynamics of the ECM remodeling process within each preclinical colitis model of colitis need to be further characterized in future studies to fully elucidate the relationship between the ECM and targeted probiotic residence time.

Studies have shown that the ECM in the inflamed GI tract is dynamic in nature and its expression, deposition, and remodeling changes in response to disease progression, severity, and degree of inflammation within the lesion¹⁸⁻²². Moreover, upregulated ECM in inflammatory lesions plays a pivotal role in downstream signaling, triggering additional inflammation. This perpetuates a challenging cycle of inflammation. While the body increases ECM protein production to structurally safeguard the area and promote healing, dysregulated and excessive deposition of these proteins leads to further inflammation and an inability to heal the lesion^{19,20,23}. Interventions aimed at regulating excessive ECM production may offer an alternative or complementary strategy for future therapeutic development. However, given the critical role of ECM in the healing process, achieving a fine-tuned balance will be essential.

6. In Figure 4J, while FN exhibits localisation, presentation (in primary Figure or supplementary) of FB and CIV data would be beneficial to assess both if faecal CFU load correlates with colonic load and variation between FB, CIV and FN and related association with phenotypic outcomes. Image quantification could also be beneficial in this context.

We agree with the reviewer on the importance of relating fecal load to colonic load and connecting that to the phenotypic outcomes. We would like to point out that **Figures 4B and 4C** most accurately quantify the fecal/colonic loads. Consistent with the fecal CFU load findings, the CFU count in the entire colon tissue also revealed a 100-fold greater retention of FN-S.b. when compared to S.b. mSA. This is further confirmed by the histochemical analyses conducted on the colon tissue (as illustrated in **Figure 4J**), which demonstrate minimal binding of S.b. mSA to the colon tissue, whereas FN-S.b. showed significantly higher levels of binding to the colon tissue.

We have added the following paragraph to **lines 370-372**:

The image analysis performed here represents a qualitative snapshot showing the enhanced retention of *S.b. FN* compared to *S.b. mSA* within the colon tissue. These findings are further supported by the quantitative evaluation of colon tissue load as described in Fig. 4C.”

7. The comparison of the acute and relapsing model provides significant insights into the mechanisms of action. This stated, more detailed explanation and rationale for the first *S.b.* gavage being undertaken at day 9 in the relapsing model as opposed to day 5 in the acute model could aid interpretation of this comparison.

We have now added a more detailed explanation and rationale for the dosing regimen.

The DSS dosing and recovery periods in the relapsing model were modeled after previous reports²⁴⁻²⁸. In the chronic model, Day 9 was chosen to begin dosing yeast as opposed to starting at Day 5 to allow for a full cycle of inflammation and recovery prior to dosing. This was to simulate human disease more closely, where a patient may not receive treatment until their second (or more) relapse of inflammation. We have included this rationale and a discussion on dosing strategies in the main text.

The following paragraphs were added:

Lines 409-420: Utilization of repeated cycles of 3-7 days of DSS administration followed by 5-14 days of recovery without DSS is a widely reported strategy to assess chronic colitis in mice²⁴⁻²⁸. Here, mice received DSS in the drinking water for 5 days followed by a recovery phase where they were placed back onto normal drinking water for 3 days. This process was repeated twice more (**Fig. 5A**). Mice received oral gavages of non-targeted (*S.b. mSA*) or ECM-targeted (*S.b. CIV* or *S.b. FN*) yeast every 3 days starting on day 9. Dosing with yeast commenced on day 9 of the study to allow for 1 complete cycle of inflammation and recovery. This is to resemble the human disease more closely, wherein a patient may not receive a treatment until after they've gone through a period(s) of inflammation, recovery, and relapse^{24,29}. Here, we combine fecal CFU concentration values from each 48- or 72-hour post-dose timepoint across the entire study to gather a high-level snapshot of CFU concentrations in the feces (**Fig. 5B**). A data plot showing all the fecal CFU concentrations across the study can be found in **Supplementary Fig. 5**.

Lines 561-566: In this study, we successfully demonstrated that the targeted *S.b. CIV* enables a significant reduction in dosing frequency, moving away from the daily reported dosing schedules^{10,30,31} to an interval of once every three days. Furthermore, we show significant recovery in both

acute and chronic DSS models of colitis with the targeted *S.b. CIV*. We posit that this platform can be further improved through stable expression of ECM-specific ligands on the yeast cell surface, thereby enabling continuous targeting capabilities with fewer doses.

Supplementary Figure 5. Fecal CFU of *S.b.* over time in a relapsing-remitting model of DSS colitis. (A) Fecal yeast concentrations at all timepoints indicated in Fig. 5 (n = 5). Bars represent mean, error shown as standard deviation, significance assessed using ordinary one-way ANOVA with Kruskal-Wallis multiple comparisons test $\alpha = 0.05$, * $p < 0.05$. (B) In-depth analysis of fecal CFU over time in the *S.b. CIV* treated group at timepoints 24 hours, 48 hours, and 72 hours post-dose. Significance was assessed using ordinary one-way ANOVA with Tukey's multiple comparisons test. No significant differences were found between the groups.

8. In all the fecal CFU counts obtained in the relapsing DSS mode the 48hrs and 72hrs post dose CFU counts vary from below detection to 10^5 CFU/g. Has this resulted from combining datasets across the time course (i.e. 48hrs Post-dose represents Day 11, Day 14, Day 17, etc.) or does this data represent a single time point with a clear dichotomy in colonisation success? The analysis and/or visualisation of this data should be modified to improve clarity and represent the data for each time point considered. Similarly, including the CFU curves, probably as supplementary, from which the data for 5C was calculated would provide a clearer indication of stability of colonisation over time.

We agree that data in Figures 5B and 5C would benefit from additional representation of CFU concentrations from each timepoint. While the current values in **Figures 5B and 5C** represent a combined dataset of CFU counts across the time points, we recognize the need for a clearer depiction. Thus, we have included all of the CFU concentration timepoints for the entirety of the chronic model in Supplementary Figure 5.

We have included an additional explanation of the combined data visualization in the main text.

Lines 417-420: Here, we combine fecal CFU concentration values from each 48- or 72-hour post-dose timepoint across the entire study to gather a high-level snapshot of CFU concentrations in the feces (**Fig. 5B**). A complete visualization of all the fecal CFU concentrations across the study can be found in **Supplementary Fig. 5.**"

9. In the relapsing DSS model, given the decreased inflammation (and therefore CIV binding opportunity) over the 25 days - was a reduction in CFU load observed as the inflammatory state was resolved?

To address the reviewer's comment we have now added the following paragraph:

Lines 505-519: In the relapsing model, we observed that within the first 24 and 48 hours after dosing, the CFU load in the feces was highest in the S.b. FN treated group. This observation reflects the trends seen in the acute DSS model (**Figure 4**). However, at later timepoints, the S.b. CIV treated group exhibited the highest fecal CFU load, consistent with data demonstrating highest Col4a1 expression in the colon (**Fig 5J**). This shift suggests changing expression levels of fibronectin to collagen IV within the colon. It also emphasizes the potentially important role that ECM remodeling plays in probiotic residence time. At the conclusion of our relapsing study, we observed a reduction in inflammatory markers (such as weight, colon length, and cytokine expression levels) in the S.b. CIV treated group. However, interestingly, there was no corresponding decrease in fecal colony-forming units. One possible explanation for this discrepancy is that although the inflammation was mitigated, it did not fully return to healthy levels. Our data, as depicted in **Fig. 5J**, revealed elevated levels of Col4a1 in the chronic DSS model. Consequently, the partial upregulation of Col4a1 may play a role in mediating prolonged retention. The spatiotemporal dynamics of the ECM remodeling process within each preclinical colitis model of colitis need to be further characterized in future studies to fully elucidate the relationship between the ECM and targeted probiotic residence time.

10. Examination of the response within the gut microbiome has the potential to provide important understandings into the capacity of S.b. CIV to modulate disease outcome. Unfortunately, the amplicon sequencing analysis has only provided minimal insight. Could this analysis be further expanded to explore key microbe dynamics over time. Variation between mice should also be clearly presented. Importantly, the relevant comparison would be S. b. mSA compared to S.b. CIV, ideally with S.b. FN and S.b. FB also included in the analysis.

The reviewer brings up an important consideration in the microbiome analysis described here. Furthermore, the role of specific changes in microbiome and how they relate to IBD progression is still not fully understood within the field. Thus, we have decided to set aside this study for future work, where we will focus on a more robust time course of microbiome changes throughout disease progression and treatment. This approach is necessary to fully understand the impact of treatment on microbiome diversity dynamics and taxonomy and such a detailed analysis is outside the scope of this manuscript.

We have added the following paragraph:

Lines 552-558: Several reports indicate the importance of the microbiome composition and diversity in disease progression and treatment response for inflammatory bowel diseases^{32,33}. However, the precise role of the microbiome in IBD progression remains incompletely understood within the field. Further research is necessary to comprehensively assess microbiome changes throughout disease

progression and in response to treatment, allowing a deeper understanding of treatment impact on microbiome diversity dynamics.

Minor points:

1. In the abstract the authors state these studies demonstrate the potential for "patient-friendly" treatments for IBD, it is unclear how this study provides this outcome.

This statement has been removed from the abstract. A further description on how oral dosage forms serve as "patient-friendly" alternatives to injection or infusion methods for IBD treatment has been added to the introduction.

Lines 49-50: Overall, these studies highlight the potential of targeted microbial therapeutics as a potential oral dosage form for the treatment of inflammatory bowel diseases.

Lines 65-69: Engineered therapeutic microbes have the potential to serve as a locally acting treatment strategy, reducing the risk of systemic side effects commonly associated with systemically administered therapeutics. Additionally, engineered microbes and probiotics are well-suited for oral dosage forms, which patients generally prefer over injections or infusion routes.

2. Line 302: Grammatical error: "as S.b. FN did not persist within the healthy colon as long in a healthy mouse"

The wording of this sentence has been changed for clarity.

Lines 333-335: Extended studies in an acute DSS model of colitis showed that the S.b. FN did not permanently engraft in the intestine (**Supplementary Fig. 3**).

2. Line 658: Grammatical error: "for 2 minutes. System then heated at 40 degrees"

This method section has been removed as the short chain fatty acid analysis has been repeated with additional controls and additional timepoints. See new methods section for SCFA analysis:

Lines 694-710:

Stimulation of short chain fatty acid production (Fig 3C-E): Overnight cultures of S.c., S.b., or S.b.-mSA were diluted into fresh CMM to reach a final OD₆₀₀=0.3. S.b. mSA cultures were centrifuged and incubated with 100 nM of biotin-anti-fibronectin, biotin-anti-fibrinogen, or biotin-anti-collagen IV (Abcam, ab6584, ab51416, or ab6581, respectively) for 30 minutes, then washed and resuspended with CMM. Cultures were incubated at 37°C for 18 hours in aerobic conditions. Cultures were then centrifuged, and aliquots of media supernatant were collected 6, 12, and 18 hours post-CMM inoculation for LC-MS analysis of SCFAs. A volume of 100 µl cell media was used for derivatization with 10 µl 2.4 M aniline and 10 µl 1.2 M N-(3-Dimethylaminopropyl)-N'-ethylcarbodiimide hydrochloride (EDC) and placed on ice for 2 hr³⁴. Acetate, propionate, and butyrate were quantified on an AB SCIEX (Foster City, CA) QTRAP 6500 system consisting of a SHIMADZU Nexera ultra high-performance liquid chromatography system coupled with a hybrid triple quadrupole and ion trap mass spectrometer. Analyst® 1.6 software was used for system control and data acquisition. Data processing and quantitation were performed with the MultiQuant 3.0 software. A straight line was fitted to the linear portion of secreted SCFA amount, before saturation was reached (0 to 18h for acetate, 0 to 12h for butyrate and propionate). The slopes of the linear regression were used to assess the secretion rates of SCFA.

Reviewer #2 (Remarks to the Author):

The manuscript by Heavey et al. describes the development of a targeted probiotic yeast platform that attaches to extracellular matrix (ECM) proteins present in inflammatory lesions of the murine GI tract. Using a biotin-streptavidin capture the authors attach antibodies specific for 3 ECM proteins. In murine models of ulcerative colitis, the engineered yeast platform binding to fibronectin demonstrated significantly longer residence time in the gut and increased abundance in the colon compared to fibrinogen and collagen IV binding strains as well as the strain not conjugated with antibodies. The fibronectin and collagen IV binding strains demonstrate benefits in short and long term DSS model. The results include increased colon length, improved colonic cytokine expression profiles, reduced histological inflammation scores, and enhanced microbiome diversity indices. The study provides an interesting approach to improving residence time and lays the groundwork for future work on improved engraftment and target specific treatment.

We thank the reviewer for their thoughtful comments in the effort to significantly improve the quality of this manuscript.

General comments:

- To avoid confusion and ensure accurate interpretation of the data presented, the authors should maintain consistency in the statistical methods used throughout the manuscript. This is especially important in Figure 4 and 5, where two different multi comparison tests are used inconsistently. Additionally, the legend should state the reference/control when a Dunnett's test is used.

All statistical tests have been repeated to maintain consistency throughout the manuscript. Reference/control information has been added to each figure legend. Dunnett's test was used when the study contained one control and Tukey's test was used when there was more than one control group.

- The authors should provide more information on the potential involvement of ECM proteins in regulating inflammation, as well as the potential anti-inflammatory effects of dosing antibodies alone. This would justify the lack of an antibodies alone control group.

As suggested by the reviewer we have included an additional paragraph to the discussion and conducted an additional study to assess the effects of the antibodies alone. In an acute DSS model, when antibodies were orally administered alone, there were no significant differences in the colon length between control and antibody-treated mice, indicating that the antibodies alone do not exert any therapeutic effects (**Supplementary Fig. 4E**). The following revisions have been made:

Lines 520-529: Studies have shown that the ECM in the inflamed GI tract is dynamic in nature and its expression, deposition, and remodeling changes in response to disease progression, severity, and degree of inflammation within the lesion¹⁸⁻²². Moreover, upregulated ECM in inflammatory lesions plays a pivotal role in downstream signaling, triggering additional inflammation. This perpetuates a challenging cycle of inflammation. While the body increases ECM protein production to structurally safeguard the area and promote healing, dysregulated and excessive deposition of these proteins leads to further inflammation and an inability to heal the lesion^{19,20,23}. Interventions aimed at regulating excessive ECM production may offer an alternative or complementary strategy for future therapeutic development. However, given the critical role of ECM in the healing process, achieving a fine-tuned balance will be essential.

Lines 358-362: To ensure that therapeutic effects were driven by *S.b.* and not the attached antibodies, we conducted an additional study by dosing the antibodies alone. In an acute DSS model, when antibodies were orally administered alone, there were no significant differences in the colon length

between control and antibody-treated mice, indicating that the antibodies only mediate binding and retention but do not exert any therapeutic effects (Supplementary Fig. 4E).

Supplementary Figure 4. Sb and Sb mSA show similar effects with no statistical differences in acute DSS colitis. ECM-targeting antibodies alone exert no significant effect. (A) Mean percent body weight of mice over the course of the study. n=5, significance indicates comparisons to average weights of DSS-only mice at each timepoint. Arrow indicates day of DSS removal and yeast dosing. **(B)** Semi-quantitative histological scores of inflammation accounting for extent of mucosal loss, hyperplasia, and erosions, n = 5. **(C)** Colon lengths following administration of engineered *S. boulardii* in acute DSS colitis model. **(D)** Representative images of hematoxylin and eosin staining of colon Swiss rolls. **(E)** Colon lengths following administration ECM-targeting antibodies in acute DSS colitis model. Bars represent mean, error shown as standard deviation, significance assessed using ordinary one-way ANOVA with Tukey’s multiple comparisons test $\alpha = ** p < 0.01$.

- It would be preferable for the authors to conduct a follow-up test to confirm that the *S.b.* mSA coating does not permanently engraft in the inflammatory induced mice model. This would help ensure the safety of the method and provide more insights into the dynamics of the colonization profile.

follow-up test has been conducted to ensure that the *S.b.* FN strain does not permanently engraft in the intestine of the DSS colitis model. Because the *S.b.* FN strain showed enhanced and prolonged retention compared to *S.b.* mSA, we chose to do the long-term colonization study with *S.b.* FN. A. This data has been included in the supplementary material (Supplementary Figure 3).

Acute DSS Colitis

Supplementary Figure 3. Long-term *S.b.* pharmacokinetic profiles in an acute model of DSS colitis. (A) Fecal CFU following administration of 10^9 *S.b.*, *S.b. mSA*, or *S.b. FN* in acute model of DSS colitis. Data are represented as mean, $n = 3$. Dotted lines represent the limit of detection and shaded regions below represent any values below the limit of detection.

Lines 333-335: Extended studies in an acute DSS model of colitis showed that the *S.b. FN* did not permanently engraft in the intestine (**Supplementary Fig. 3**).

Major comments:

Line 52. Engineered probiotics and LBPs are not the same thing. In addition, an LBP doesn't necessarily need to be a probiotic microbe. The term "engineered live biotherapeutic products (eLBPs)" is a commonly used term that could be adopted for clarity. The authors should also use consistent terminology throughout the manuscript in response to "engineered therapeutic microbes" on line 65.

The terminology "live biotherapeutic products" has been removed from **Line 52** to reduce confusion. it now reads: "Engineered probiotic microbes represent an emerging pharmaceutical strategy with the ability to modulate the gut microbiome".

Line 113. The authors should acknowledge that removing fungal infections with anti-fungal agents is not always a quick and easy solution, and that some anti-fungal agents can also impact gut bacteria, as seen in mice (doi: 10.1128/AAC.02552-20). This statement should be modified to reduce any confusion among readers.

This statement has been modified to include these considerations.

Lines 116-117: amenable to removal through anti-fungal administration with reduced impact on the gut microbiome as compared to broad spectrum antibiotics^{35,36}.

Line 115-116: The sentence feels a bit redundant to the previous paragraph.

This sentence has been removed to avoid redundancy.

Line 119-121. The authors should clarify that the statistical comparisons in the figures have only been compared to DSS-only. Likewise in similar cases where the antibody coated S.b. mSA is claimed to improve the state of the disease.

All statistical tests have been updated and comparisons are noted in the figure legends. Comparisons to S.b. mSA have also been included in the figures to provide further clarity in the differences between groups.

Line 313-314. The authors should use the correct multi comparison test to confirm that there is no significant difference between S.b. FN and non-DSS healthy mice. As now a Dunnett's test is used according to Figure 4E.

Statistical tests have been updated in the figure legend to provide further clarity. Figure legend for Figure 3 in **Lines 298-313** now reads:

Figure 3. Engineered *S. boulardii* retain probiotic mechanisms of action *in vitro*. (A) Schematic of probiotic mechanisms of action elicited by *S. boulardii*. (B) Percent viability after 4 h incubations of control and engineered yeast in media at pH 2.5, pH 4.0, or containing 0.3% or 0.6% OxGall bile salts (OxG). Data are shown as mean \pm SD, n=3. Significance was determined using ordinary one-way ANOVA with Tukey's multiple comparisons test. (C) Secretion of acetate, propionate, and butyrate from cultures of either *S.b.*, *S.b. mSA*, *S.b. FN*, *S.b. FB*, and *S.b. CIV*, or blank media, n = 4. Solid line: simple regression of the linear portion before saturation (0 to 18 h for acetate, 0 to 12 h for butyrate and propionate). (D) Secretion rate of acetate, propionate, and butyrate from cultures of either *S.b.*, *S.b. mSA*, *S.b. FN*, *S.b. FB*, and *S.b. CIV*, or blank media calculated from linear regression in (C), n = 4. (E) Concentration of murine interleukin-10 (IL-10) in the cell culture supernatant following an 18 h co-incubation of murine bone marrow-derived dendritic cells with controls and engineered yeast, n = 3. (F). IL-8 concentrations in stimulated HT-29 epithelial co-cultured with probiotic yeast strains. Data are shown as mean \pm SD. Significance was determined using ordinary one-way ANOVA with Tukey's for panels B, D, E, and F. For panels B, E, and F: black asterisks indicate significance against PBS and grey asterisks indicate significance against *S.c.*, $\alpha = 0.05$, * $p < 0.05$, ** $p < 0.01$, *** $p < 0.001$, **** $p < 0.0001$.

Line 336. The authors should also confirm whether Figure 4F was corrected for multiple comparisons of genes.

Statistical tests between groups for each gene were independently performed then represented in a single heatmap in Figure 4F. This information has been included in the figure legend.

Lines 386-390: (F) Mean relative expression of pro-inflammatory (*Tnfa*, *Ifn γ* , *Il6*) and anti-inflammatory (*Tgfb*, *Il10*) cytokines in distal colon tissue compared to healthy controls, n = 5. Statistical tests between groups for each gene were independently performed then represented in a single heatmap. Black asterisks indicate significance against DSS and grey asterisks indicate significance against *S.b. mSA*.

Line 358. The authors should include a statistical test for Figure 4I.

A statistical test has been performed for Figure 4I. No significant differences in expression were seen between test groups. This information has been included in the main text to provide further clarity.

Lines 352-355: Although not statistically significant from the other groups, fibronectin expression was the most upregulated in DSS-only colons vs. healthy colon tissue, indicating a potential trend between ECM expression and targeted probiotic retention time.

Line 366-367. The authors should consider providing a reference for the cycle of 5 days DSS and 3 days recovery. Or elaborate further on this specific dosing regimen.

A reference to the cycling DSS dosing regimen has been added.

Lines 409-413: Utilization of repeated cycles of 3-7 days of DSS administration followed by 5-14 days of recovery without DSS is a widely reported strategy to assess chronic colitis in mice²⁴⁻²⁸. Here, mice received DSS in the drinking water for 5 days followed by a recovery phase where they were placed back onto normal drinking water for 3 days. This process was repeated twice more (**Fig. 5A**).

Line 367. The magnitude change is closer to 1000-fold than 100-fold in Figure 5D.

This magnitude has been changed to 1000-fold in the main text,

Line 427: Viable *S.b. CIV* yeast were also detected in higher abundance in the colon tissue at the study endpoint, where mice treated with *S.b. CIV* had up to 1000-fold higher concentrations of tissue-associated *S.b.* compared to other treatment groups (**Fig. 5D**).

Line 385. The authors should clarify whether the effect observed is significant or not. Similar described in the earlier paragraph for Figure 4.

Information regarding significant differences between gene expression levels has been added to this section.

Lines 462-465: (G) Mean relative expression of pro-inflammatory (*Tnfa*, *Ifng*, *Il6*) and anti-inflammatory (*Tgfb*, *Il10*) cytokines in distal colon tissue compared to healthy controls, n = 5. Black asterisks indicate significance against DSS and grey asterisks indicate significance *S.b.* mSA.

Line 387-388. The authors should consider providing more quantitative information, similar to the previous section for Figure 4.

Additional quantitative information has been added to this section.

Line 418. The authors should specify the statistical test used for Figure 5G and Figure 5J.

Statistical test information has been further specified in the figure legend.

Line 437. The authors should clarify whether the observed effect is due to Sb or prolonged Sb colonisation. The authors should also elaborate why Sb control was not included.

Due to the potentially significant, yet limited information gathered in the 16S sequencing study, we have decided to remove Figure 6 from the manuscript. We believe further characterization with more controls and timepoints is necessary to draw confident conclusions regarding the role of the microbiome in these studies. Furthermore, the precise role of specific members of the microbiome in IBD progression remains incompletely understood within the field. As a result, we have decided to set aside this study for future work, where we will focus on a more robust time course of microbiome changes throughout disease progression and treatment. This approach is essential to fully understand the impact of treatment on microbiome diversity and compositional dynamics – a topic that lies beyond the scope of this manuscript. The primary objective of this manuscript is to highlight

the novel engineering strategy used for modifying *S.b.* and its potential advantages for the treatment of colitis.

To address the reviewers' comment, we added the following paragraph to the discussion section: **Lines 552-558.** Several reports have indicated the importance of the microbiome composition and diversity in disease progression and treatment response for inflammatory bowel diseases^{32,33,37}. However, the precise role of the microbiome in IBD progression remains incompletely understood within the field. Further research is necessary to comprehensively assess microbiome changes throughout disease progression and in response to treatment, allowing a deeper understanding of treatment impact on microbiome diversity dynamics.

Line 462-465. The authors should experimentally show that coated *S.b.* mSA is cleared out to strengthen the safety of this binding method.

Clearance of coated *S.b.* mSA in DSS mice can now be found in the supplemental material (**Supplementary Figure 3**).

Minor comments:

Line 45. The authors could consider including more quantitative data on the prolonged yeast colonisation in order to provide a clearer understanding of the extent of the effect.

Further quantitative information has been added.

Line 45-46: This approach enabled an additional 24-48 hours of probiotic gut residence time compared to controls and 100-fold increased *S.b.* concentrations within the colon in preclinical models of murine ulcerative colitis.

Line 75: The authors abbreviation of "Saccharomyces boulardii (*S.b.*)" should be "(*S.b.*)" to be consistent with later abbreviation style.

All instances of abbreviation for *Saccharomyces boulardii* have been changed to "*S.b.*".

Line 104. *Saccharomyces boulardii* has already been abbreviated and therefore not necessary to do it again.

Abbreviation repetition has been removed.

Line 159. For better figure aesthetic authors should consider using consistent colour coding for the linearised plasmid in A and B; and should also state the terminator used.

Terminator information has been added to the figure. Colors have been changed to maintain consistency.

Line 159. The authors should consider providing the sequence used in the supplementary would improve knowledge sharing and reproducibility among the scientific communities.

The plasmid sequence has been added to the supplementary material (**Supplementary Table 1**).

Line 199. "*S.b.*" should be corrected to "*S.b.* mSA."

Line 201 "*S.b.*" has been corrected to "*S.b.* mSA"

Line 214. The authors should include a legend for Figure 2B, similar to Figure 3B.

A figure legend for Figure 2B has been added.

Lines 219-224: (B) Quantification of attached targeted *S.b.* (colored bars) or non-targeted *S.b.* (grey bars) on fibronectin-coated (FN blue), fibrinogen-coated (FB purple), or collagen IV-coated (CIV red) well plates at varying seeding densities. Images were quantified using ImageJ, n=3 wells per condition. Bars represent mean, error shown as standard deviation, significance assessed using ordinary one-way ANOVA with Šídák's multiple comparisons test, n=3 per condition.

Line 272. The statistical comparison lines should be aligned to improve readability.

Statistical comparison lines have been aligned in Figure 3.

Line 336. The authors should write "Log₁₀" instead of "log," and should clearly indicate what the dotted and shadow areas represent in the figure and text legend.

All instances of "Log" have been changed to "Log₁₀". The dotted lines represent the limit of detection for the corresponding assay. Descriptions of this have been added to each figure legend.

Line 530, 607, 624. The authors should use consistent terminology, such as "S.b. mSA" instead of "S.b. -mSA."

All instances where a hyphen is used (S.b.-mSA) have been changed (S.b. mSA).

Reference:

1. Calvigioni, M. *et al.* HPLC-MS-MS quantification of short-chain fatty acids actively secreted by probiotic strains. *Front. Microbiol.* **14**, 1124144 (2023).
2. Sougioultzis, S. *et al.* *Saccharomyces boulardii* produces a soluble anti-inflammatory factor that inhibits NF-kappaB-mediated IL-8 gene expression. *Biochem. Biophys. Res. Commun.* **343**, (2006).
3. Park, J. Y. *et al.* NF-kappaB-dependency and consequent regulation of IL-8 in echinomycin-induced apoptosis of HT-29 colon cancer cells. *Cell Biol. Int.* **32**, 1207–1214 (2008).
4. Thomas, S. *et al.* *Saccharomyces boulardii* inhibits lipopolysaccharide-induced activation of human dendritic cells and T cell proliferation. *Clin. Exp. Immunol.* **156**, 78–87 (2009).
5. Cristofori, F. *et al.* Anti-Inflammatory and Immunomodulatory Effects of Probiotics in Gut Inflammation: A Door to the Body. *Front. Immunol.* **12**, 578386 (2021).
6. Krawiec, P., Pawłowska-Kamieniak, A. & Pac-Kożuchowska, E. Interleukin 10 and interleukin 10 receptor in paediatric inflammatory bowel disease: from bench to bedside lesson. *J. Inflamm.* **18**, 13 (2021).
7. Meng, D., Liang, L. & Guo, X. Serum interleukin-10 level in patients with inflammatory bowel disease: A meta-analysis. *Eur. J. Inflamm.* **17**, 205873921984340 (2019).
8. Keubler, L. M., Buettner, M., Häger, C. & Bleich, A. A Multihit Model: Colitis Lessons from the Interleukin-10-deficient Mouse. *Inflamm. Bowel Dis.* **21**, 1967–1975 (2015).
9. Wang, B. *et al.* *Saccharomyces boulardii* attenuates inflammatory response induced by *Clostridium perfringens* via TLR4/TLR15-MyD8 pathway in HD11 avian macrophages. *Poult. Sci.* **99**, 5356–5365 (2020).
10. Gao, H. *et al.* *Saccharomyces boulardii* Ameliorates Dextran Sulfate Sodium-Induced Ulcerative Colitis in Mice by Regulating NF-κB and Nrf2 Signaling Pathways. *Oxid. Med. Cell. Longev.* **2021**, 1622375 (2021).
11. Rahman, M. M. & McFadden, G. Modulation of NF-κB signalling by microbial pathogens. *Nat. Rev. Microbiol.* **9**, 291–306 (2011).
12. Guarino, A., Lo Vecchio, A. & Canani, R. B. Probiotics as prevention and treatment for diarrhea. *Curr. Opin. Gastroenterol.* **25**, 18–23 (2009).
13. Elliott, C. L., Allport, V. C., Loudon, J. A., Wu, G. D. & Bennett, P. R. Nuclear factor-kappa B is essential for up-regulation of interleukin-8 expression in human amnion and cervical epithelial cells. *Mol. Hum. Reprod.* **7**, 787–790 (2001).
14. Lee, S. K., Kim, Y. W., Chi, S.-G., Joo, Y.-S. & Kim, H. J. The Effect of *Saccharomyces boulardii* on Human Colon Cells and Inflammation in Rats with Trinitrobenzene Sulfonic Acid-Induced Colitis. *Dig. Dis. Sci.* **54**, 255–263 (2009).
15. Liu, T., Zhang, L., Joo, D. & Sun, S.-C. NF-κB signaling in inflammation. *Signal Transduct. Target. Ther.* **2**, 17023 (2017).
16. Kelesidis, T. & Pothoulakis, C. Efficacy and safety of the probiotic *Saccharomyces boulardii* for the prevention and therapy of gastrointestinal disorders. *Ther. Adv. Gastroenterol.* **5**, 111–125 (2012).
17. Pais, P., Almeida, V., Yilmaz, M. & Teixeira, M. C. *Saccharomyces boulardii*: What Makes It Tick as Successful Probiotic? *J. Fungi* **6**, 78 (2020).
18. Mortensen, Jh. *et al.* The intestinal tissue homeostasis – the role of extracellular matrix remodeling in inflammatory bowel disease. *Expert Rev. Gastroenterol. Hepatol.* **13**, 977–993 (2019).
19. Derkacz, A., Olczyk, P., Olczyk, K. & Komosinska-Vassev, K. The Role of Extracellular Matrix Components in Inflammatory Bowel Diseases. *J. Clin. Med.* **10**, 1122 (2021).
20. Petrey, A. C. & de la Motte, C. A. The extracellular matrix in IBD: a dynamic mediator of inflammation. *Curr. Opin. Gastroenterol.* **33**, 234–238 (2017).
21. Golusda, L., Kühl, A. A., Siegmund, B. & Paclik, D. Extracellular Matrix Components as Diagnostic Tools in Inflammatory Bowel Disease. *Biology* **10**, 1024 (2021).

22. Mortensen, J. H. *et al.* Ulcerative colitis, Crohn's disease, and irritable bowel syndrome have different profiles of extracellular matrix turnover, which also reflects disease activity in Crohn's disease. *PLoS One* **12**, e0185855 (2017).
23. Koelink, P. J. *et al.* Collagen degradation and neutrophilic infiltration: a vicious circle in inflammatory bowel disease. *Gut* **63**, 578–587 (2014).
24. Peters, L. A. *et al.* A temporal classifier predicts histopathology state and parses acute-chronic phasing in inflammatory bowel disease patients. *Commun. Biol.* **6**, 95 (2023).
25. Kwon, J., Lee, C., Heo, S., Kim, B. & Hyun, C.-K. DSS-induced colitis is associated with adipose tissue dysfunction and disrupted hepatic lipid metabolism leading to hepatosteatosis and dyslipidemia in mice. *Sci. Rep.* **11**, 5283 (2021).
26. Breyngaert, C. *et al.* Unique Gene Expression and MR T2 Relaxometry Patterns Define Chronic Murine Dextran Sodium Sulphate Colitis as a Model for Connective Tissue Changes in Human Crohn's Disease. *PLoS ONE* **8**, e68876 (2013).
27. Ben-Ami Shor, D. *et al.* Immunomodulation of Murine Chronic DSS-Induced Colitis by Tuftsin–Phosphorylcholine. *J. Clin. Med.* **9**, 65 (2019).
28. Chassaing, B., Aitken, J. D., Malleshappa, M. & Vijay-Kumar, M. Dextran Sulfate Sodium (DSS)-Induced Colitis in Mice. *Curr. Protoc. Immunol.* **104**, (2014).
29. Magro, F. *et al.* Third European Evidence-based Consensus on Diagnosis and Management of Ulcerative Colitis. Part 1: Definitions, Diagnosis, Extra-intestinal Manifestations, Pregnancy, Cancer Surveillance, Surgery, and Ileo-anal Pouch Disorders. *J. Crohns Colitis* **11**, 649–670 (2017).
30. Li, B. *et al.* *Saccharomyces boulardii* alleviates DSS-induced intestinal barrier dysfunction and inflammation in humanized mice. *Food Funct.* **13**, 102–112 (2022).
31. Wang, C. *et al.* *Saccharomyces boulardii* alleviates ulcerative colitis carcinogenesis in mice by reducing TNF- α and IL-6 levels and functions and by rebalancing intestinal microbiota. *BMC Microbiol.* **19**, 246 (2019).
32. Lee, M. & Chang, E. B. Inflammatory Bowel Diseases (IBD) and the Microbiome—Searching the Crime Scene for Clues. *Gastroenterology* **160**, 524–537 (2021).
33. Caruso, R., Lo, B. C. & Núñez, G. Host–microbiota interactions in inflammatory bowel disease. *Nat. Rev. Immunol.* **20**, 411–426 (2020).
34. Bihan, D. G. *et al.* Method for absolute quantification of short chain fatty acids via reverse phase chromatography mass spectrometry. *PLOS ONE* **17**, e0267093 (2022).
35. Jiang, T. T. *et al.* Commensal Fungi Recapitulate the Protective Benefits of Intestinal Bacteria. *Cell Host Microbe* **22**, 809-816.e4 (2017).
36. Ozdemir, T., Fedorec, A. J. H., Danino, T. & Barnes, C. P. Synthetic Biology and Engineered Live Biotherapeutics: Toward Increasing System Complexity. *Cell Syst.* **7**, 5–16 (2018).
37. Lawal, S. A., Voisin, A., Olof, H., Bording-Jorgensen, M. & Armstrong, H. Diversity of the microbiota communities found in the various regions of the intestinal tract in healthy individuals and inflammatory bowel diseases. *Front. Immunol.* **14**, 1242242 (2023).

REVIEWERS' COMMENTS

Reviewer #1 (Remarks to the Author):

The authors have done an excellent job addressing the comments raised and they should be congratulated for completing this interesting and insightful study.

I support publication of the revised manuscript.

Reviewer #2 (Remarks to the Author):

The author has revised the text and incorporated new, improved sections, which marks a positive step towards enhancing the manuscript. However, additional modifications are needed to fully meet the criteria for publication. Further refinement and clarification of certain points will be essential to satisfy the publication standards. To improve the text for your manuscript review comment:

- Ensure consistency in italicizing microbial strains throughout the manuscript.
- Align figure panels and labels properly.
- Lines 113-115 require citations for statements on "proper modification" and "auxotrophic biocontainment strategy." Please add relevant references.
- Line 136: Introduce figures in the order they appear (i.e., A, B, C, D...).
- Figure 1, please include the terminator used (terminator downstream of your genes, state the name also in supplementary information)
- Line 189: Consider if six references are necessary, keeping in mind Nature Communications' guideline of up to 70 references.
- For Figure 2, adjust the color scheme for mSA and FN Ab in Figure 2E for consistency with other sections.
- Line 264: Clarify the unit or meaning of the value "1.2".

- Line 265: Reevaluate the statement regarding the effects on butyrate, propionate, and acetate, considering significant differences observed. A 50% increase in butyrate cannot be deemed minimal. It may be advisable to omit this sentence for accuracy.
- Line 268: Abbreviate "bone marrow dendritic cells" upon first mention to avoid confusion with BMDC.
- Lines 272-273 lacks a reference.
- Line 285: Specify the TNF alpha concentration used.
- Specify the species used for the assay in Figure 3's legend.
- For Figure 3F, clarify if PBS received TNF alpha treatment and include information on the cell viability. Additionally, the inclusion of a healthy control for comparison is crucial to contextualize the results.
- Lines 307 – 309: Provide details on LPS and TNFa concentration and the number of live cells used
- Lines 334-335: Clarify the distinction between this study's findings and those depicted in Figure 4, noting the significant variation in colonization levels. Additionally, modify the assertion of "did not" occur to acknowledge the limitations of the detection method used. Suggest that the phenomenon is "below the limit of detection" to accurately reflect the sensitivity of the measurement approach employed.
- Lines 341 – 342: This is a very wordy sentence "measurements had no statistical differences to that of the non-DSS healthy mice." Consider rewrite with more precise language, example "insignificant to the non-DSS healthy mice".
- Line 492: Ensure the correct abbreviation is used; it should be FB if previously mentioned as such.
- Line 514: Maintain consistency with the abbreviation "CFU" throughout the text.
- Line 576: "S." is not necessary to write out.
- Line 578. Remove "-" in "-mSA"
- Line 581: Is the YNB without amino acids and as
- Line 585: Specify how the low pH was accomplished.
- Line 595: How was the anaerobic condition achieved.
- Line 599: Add space between "8-" and "to".
- Line 672: Remove "-" in "-mSA"
- Line 695: Remove "-" in "-mSA"

This condensed feedback aims to clarify your manuscript for scientific publication, focusing on consistency, citation adequacy, figure presentation, and precise language.

REVIEWERS' COMMENTS

Reviewer #1 (Remarks to the Author):

The authors have done an excellent job addressing the comments raised and they should be congratulated for completing this interesting and insightful study.

I support publication of the revised manuscript.

We would like to thank the reviewer for the time dedicated to reviewing and improving this manuscript.

Reviewer #2 (Remarks to the Author):

The author has revised the text and incorporated new, improved sections, which marks a positive step towards enhancing the manuscript. However, additional modifications are needed to fully meet the criteria for publication. Further refinement and clarification of certain points will be essential to satisfy the publication standards. To improve the text for your manuscript review comment:

We thank the reviewer for their attention to detail in making this manuscript polished and consistent.

- Ensure consistency in italicizing microbial strains throughout the manuscript.

Names for microbial strains have been checked throughout the manuscript and figure panels and ensured to be italicized.

- Align figure panels and labels properly.

All figures have been checked for alignment.

- Lines 113-115 require citations for statements on "proper modification" and "auxotrophic biocontainment strategy." Please add relevant references.

References have been added regarding these strategies.

- Line 136: Introduce figures in the order they appear (i.e., A, B, C, D...).

The first reference to Fig 1F was unnecessary and it has been removed. The paragraph has been edited to introduce each of the subpanels in the text as they appear in the figure.

- Figure 1, please include the terminator used (terminator downstream of your genes, state the name also in supplementary information)

The figure and supplementary information have been edited to include the names of the terminators used.

- Line 189: Consider if six references are necessary, keeping in mind Nature Communications' guideline of up to 70 references.

These references are all necessary, as they contain the original transcriptional data from 7 datasets that we pooled for our transcriptional analysis in Fig 1F. See lines 681-682 in methods.

- For Figure 2, adjust the color scheme for mSA and FN Ab in Figure 2E for consistency with other sections.

The colors in Figures 2D and 2E have been edited to ensure consistency with other sections.

- Line 264: Clarify the unit or meaning of the value "1.2".

The "1.2" value here is a fold change. This has been edited to reflect that.

- Line 265: Reevaluate the statement regarding the effects on butyrate, propionate, and acetate, considering significant differences observed. A 50% increase in butyrate cannot be deemed minimal. It may be advisable to omit this sentence for accuracy.

We have removed the word "minimal" and edited the final sentence to read "Overall, our findings suggest that genetic modifications and the presence of antibodies did not reduce the ability of engineered *S.b.* to secrete short-chain fatty acids."

- Line 268: Abbreviate "bone marrow dendritic cells" upon first mention to avoid confusion with BMDC.

An abbreviation on upon the first mention has been added.

- Lines 272-273 lacks a reference.

References 71-73 (next sentence) cover the statement in these lines. These references can be included a second time during copy editing, if the journal prefers this instead of citing them once at the end of the material.

- Line 285: Specify the TNF alpha concentration used.

TNFa concentration has been added here, in addition to its location in the methods section.

- Specify the species used for the assay in Figure 3's legend.

We indicate that 3E is murine BMDC and 3F is HT-29 epithelial cells. We now include in the legend that HT-29 are human epithelial cells.

- For Figure 3F, clarify if PBS received TNF alpha treatment and include information on the cell viability. Additionally, the inclusion of a healthy control for comparison is crucial to contextualize the results.

In figure 3F the PBS group did receive TNF α stimulation. This has been clarified in the main text, See lines 243.

It is well established that TNF α induces IL-8 in epithelial cells, including HT29, thus we repeated this established assay. Although the reviewer is correct that a non-TNF α treated control is important for assessing the effects of TNF α stimulation, our purpose was to assess the ability of the engineered yeast strains to reduce these inflammatory effects in relation to the unmodified yeast already known to reduce nfkb and IL-8. Including a non-TNF α treated control will not change the conclusions.

- Lines 307 – 309: Provide details on LPS and TNF α concentration and the number of live cells used.

We have included the concentration of LPS and TNF α to the figure legend, in addition to its inclusion in the methods section. The number of BMDC and yeast cells used in this assay are now included in the main text, See line 314.

- Lines 334-335: Clarify the distinction between this study's findings and those depicted in Figure 4, noting the significant variation in colonization levels. Additionally, modify the assertion of "did not" occur to acknowledge the limitations of the detection method used. Suggest that the phenomenon is "below the limit of detection" to accurately reflect the sensitivity of the measurement approach employed.

This sentence has been edited to include these clarifications to read: "In an acute model of DSS colitis wherein mice received DSS in the drinking water for 5 days, dosed with *S.b. FN*, followed by an extended recovery period of 14 days; *S.b. FN* was not detectable in the feces 48 hours through end of the study at day 14 (Supplementary Fig. 3). This indicates that after *S.b. FN* is cleared from the intestines, there is no permanent engraftment detected."

- Lines 341 – 342: This is a very wordy sentence "measurements had no statistical differences to that of the non-DSS healthy mice." Consider rewrite with more precise language, example "insignificant to the non-DSS healthy mice".

This sentence has been edited to remove wordiness.

"Mice treated with *S.b. FN*, showed the most significant increase in colon length and these measurements were insignificant from those of the non-DSS healthy mice."

Point-by-Point Response

- Line 492: Ensure the correct abbreviation is used; it should be FB if previously mentioned as such.

This has been changed to “FB”.

- Line 514: Maintain consistency with the abbreviation "CFU" throughout the text.

This has been changed to “CFU”.

- Line 576: “S.” is not necessary to write out.

This has been edited to remove the written out “*Saccharomyces*”.

- Line 578. Remove “-“ in “-mSA”

The hyphen has been removed here.

- Line 581: Is the YNB without amino acids and as

This is yeast nitrogen base (YNB) without amino acids. This detail has been included.

- Line 585: Specify how the low pH was accomplished.

Low pH was accomplished by adding concentrated hydrochloric acid to the media dropwise until the desired pH was reached.

- Line 595: How was the anaerobic condition achieved.

This section has been removed as it no longer pertains to the updated Figure 3.

- Line 599: Add space between “8-“ and “to”.

A space has been added here.

- Line 672: Remove “-“ in “-mSA”

The hyphen has been removed here.

- Line 695: Remove “-“ in “-mSA”

The hyphen has been removed here.

This condensed feedback aims to clarify your manuscript for scientific publication, focusing on consistency, citation adequacy, figure presentation, and precise language.